# Simultaneous adsorption of ammonia nitrogen and phosphate on electro-assisted magnesium/aluminum-loaded sludge-based biochar and its utilization as a plant fertilizer

Qi Wang[‡], Chu-Ya Wang[‡]*, Heng-Deng Zhou, Dong-Xin Xue, Xiao-Lu Xiong, Guangcan Zhu

School of Energy and Environment, Southeast University, Nanjing, China

‡ QW and C-YW are contributed equally to this work as co first authors.
* wang-cy@seu.edu.cn

## Abstract

Herein, Mg/Al-loaded sludge-based biochar was prepared via electro-assisted impregnation. The structure and chemical analysis of modified sludge-based biochar (MgSBC-0.5 (@Al) showed that the material was loaded with MgO and $Al_2O_3$. The specific surface area of MgSBC-0.5(@Al) was 11.27 times higher than that of unmodified sludge-based biochar (SBC). The simultaneous adsorption performance of MgSBC-0.5(@Al for ammonia nitrogen ($NH_4^+$–N) and phosphate phosphorus ($PO_4^{3-}$–P) was studied. The maximum adsorption capacities of MgSBC-0.5(@Al for $NH_4^+$–N and $PO_4^{3-}$–P at 298 K were 65.19 and 92.10 mg·g$^{-1}$, respectively, 4.45 and 6.28 times higher than those of SBC. The external and internal elemental compositions of the modified and unmodified biochar specimens were quantitatively characterized using inductively coupled plasma mass spectrometry, X-ray photoelectron spectroscopy, and X-ray fluorescence spectrometry. The results emphasized the importance of Mg-loading for $NH_4^+$–N and $PO_4^{3-}$–P capture. MgO was mainly loaded on the surface of biochar, enabling adsorption through chemical reactions. Analysis showed that the adsorption of $NH_4^+$–N and $PO_4^{3-}$–P on the modified biochar proceeded simultaneously through multiple mechanisms. Particularly, the adsorption of $NH_4^+$–N and $PO_4^{3-}$–P occurred through the precipitation of struvite and physical adsorption, with $PO_4^{3-}$–P also adsorbed through the formation of $MgHPO_4$ and $CaHPO_4$. Other data indicated that Al, Ca, and Fe had a trapping effect on the adsorbate. Importantly, the biochar after adsorption could be used as a soil amendment.

**Data Availability Statement:** All relevant data are within the manuscript and its Supporting Information files.

## Introduction

With the recent surge in industrialization, the size of sewage treatment facilities has rapidly increased. The production of sludge, as a by-product of sewage treatment plants, has also been gradually increasing. In Europe, the annual production of sludge can reach 10 million tons (in

**Funding:** This study was financially supported by the Natural Science Foundation of Jiangsu Province in the form of grants received by C-YW (BK20211047) and GZ (BK20220038). No additional external funding was received for this study.

**Competing interests:** NO authors have competing interests.

dry weight) [1]. The degradation rate of organic matter in sludge after anaerobic fermentation is only 40%–60%, with a considerable amount of organic matter remaining in the fermentation residue [2]. The discharge of sewage sludge containing nitrogen and phosphorus detrimentally affects water quality, leading to the eutrophication of water bodies [3]. Therefore, there is a pressing need to develop efficient and cost-effective methods for treating wastewater and sludge.

Biochar (BC) is a porous, loose, aromatic solid product formed through the pyrolysis of biomass under limited oxygen conditions. BC has a certain amount of surface functional groups, large specific surface area, wide availability of required raw materials, and low cost. Additionally, BC can adsorb pollutants from water [4]. Agricultural waste (such as straw, sawdust, sugarcane bagasse, and rice bran) and animal manure can be used as raw materials for the production of BC [5]. Some studies have shown that BC prepared from sludge exhibits strong adsorption capacity for organic and metal pollutants. Pyrolysis substantially reduces the ecological toxicity of heavy metals in sludge BC, reducing the environmental risks of its application [6]. Compared to BCs based on other types of biomass, sludge BC can be used to cost-effectively adsorb N and P. Therefore, sludge BC has unique advantages over other BC materials in practical applications [7]. However, the functionality and adsorption capacity of BC prepared via conventional pyrolysis are limited, making it unsuitable for pollutant adsorption. Consequently, BC is chemically modified, including impregnation modification with single metals (Fe, Mg, etc.) [8–10] and pairs of metals (Mg/Al, Ca/Mg, etc.) [11,12]. Among the modified BCs, magnesium salt–modified biochar has high surface activity, anion fixation ability, and ion exchange ability. Magnesium-modified BC (Mg–BC) has been previously studied as a potential adsorbent for N and P. Mg–BC is prepared mainly through impregnation carbonization and carbonization impregnation. The modification of BC with Mg has been performed by soaking it in $MgCl_2$ solution. However, the immersion time is long, usually more than 2 h [13]. Therefore, some researchers have proposed electro-assisted modification. Mg/Al bilayer metal–modified BC was prepared via an electro-assisted method using $MgCl_2$ solution electrolyte and Al electrode, with an impregnation time of only 5 min. The formed compounds (MgO, spinel $MgAl_2O_4$, AlOOH, and $Al_2O_3$) evenly covered the surface of BC, exhibiting a highly organized and well-defined structure, resulting in improved $PO_4^{3-}$–P adsorption capability of BC.

However, such a modification produces toxic $Cl_2$ gas and was employed only in the studies on the adsorption of $PO_4^{3-}$–P from sewage. However, in sewage, $NH_4^+$–N may co-exist with $PO_4^{3-}$–P. Moreover, the BC material used in the above study was prepared from seaweed, which is more expensive than sludge. Importantly, $NH_4^+$–N may accumulate in the sewage treatment plant owing to anaerobic sludge digestion. However, $PO_4^{3-}$–P and $NH_4^+$–N coexisting in solution may be removed using $Mg^{2+}$. Therefore, sludge-based BC (SBC) has unique advantages over other BC materials in practical applications.

Therefore, the present study aimed to prepare and characterize a bimetallic-modified SBC, with modification performed using Mg salt solution as an electrolyte in an electro-assisted system. Additionally, simultaneous adsorption capabilities of the prepared material for $NH_4^+$–N and $PO_4^{3-}$–P in wastewater were determined. The results show that sludge can be modified within 10 min and pyrolyzed to obtain functionalized sludge-based BC (Mg/Al assembled composite) with a high surface area. The obtained results provide a valuable reference for the preparation of modified SBCs for the adsorption of $NH_4^+$–N and $PO_4^{3-}$–P. Additionally, the applicability of saturated adsorbents as soil amendments was also explored. Overall, the use of modified SBC is a green and novel method for recovering $NH_4^+$–N and $PO_4^{3-}$–P from sewage sludge.

## Materials and methods

### Preparation of materials

After anaerobic fermentation, the sludge was dried in an oven at 105°C to remove moisture content, ground to a fine powder (smaller than 200 mesh), and stored in a dry place.

Mg/Al-loaded SBC (MgSBC-0.5(@Al)) was prepared by immersing 2 g of dry sludge powder (named SP) in 0.5 mol·$L^{-1}$ magnesium acetate solution (100 mL). Subsequently, the pH of the solution was adjusted to 3.0 using 0.25 M HCl and NaOH. Electro-assisted modification was performed using a power supply to apply the appropriate current density (fixed at 93.96 mA $cm^{-2}$; programmable DC power supply, ODA, Korea). The anode and cathode rods of the electrochemical modification experimental device were both Al electrode, and a saturated calomel reference electrode was used. Aluminum electrodes had an effective surface area of 0.07065 $cm^2$, and the separation between the electrodes was adjusted to 3 cm. The electro-assisted modification was conducted under constant stirring at 120 rpm for 10 min. The modified sludge was dried at 65°C, pyrolyzed at 500°C (heating rate: 5°C·$min^{-1}$) for 2 h in a nitrogen ($N_2$) environment, ground to a fineness of 200 mesh, and encapsulated in a container.

MgSBC-0.1(@Al), MgSBC-0.25(@Al), and MgSBC-1(@Al) were prepared following the same procedure as MgSBC-0.5(@Al but using different magnesium acetate concentrations during electro-assisted modification-0.1, 0.25, and 1 mol/L, respectively. MgSBC-0.5(@Al) was prepared by soaking SP in 0.5 mol·$L^{-1}$ magnesium acetate and stirring at 120 rpm for 6 h. The drying, pyrolysis, and subsequent procedures were the same as in the preparation of MgSBC-0.5(@Al). The unmodified SBC specimen (named SBC) was prepared via thermal decomposition at 500°C (heating rate: 5°C·$min^{-1}$) in $N_2$ environment for 2 h.

All reagents were obtained from Aladdin Industrial Company (Shanghai, China). Solutions containing $NH_4^+$–N (measured in N units) and $PO_4^{3-}$–P at concentrations of 1000 mg·$L^{-1}$ were prepared by dissolving appropriate amounts of $NH_4Cl$ and $KH_2PO_4$ in deionized water and diluting them to 1000 mL in a volumetric flask. Subsequently, the prepared stock solution was diluted to the desired concentration.

### Batch adsorption experiments

The effects of initial pH, concentration, contact time, and presence of other ions on the adsorption properties of the prepared materials were determined using the isotherm adsorption method. Adsorption solution containing 50 mg·$L^{-1}$ of $PO_4^{3-}$–P and 100 mg·$L^{-1}$ of $NH_4^+$–N was used in this and subsequent experiments. Modified BC (0.02 g) was added to 30 mL of the adsorption solution, the pH was adjusted to 7, and the mixture was shaken at a speed of 150 rpm. Then, a syringe filter with a pore diameter of 0.45 μM was used to separate the mixture. The quantities of $NH_4^+$–N and $PO_4^{3-}$–P adsorbed were calculated using Eq (1):

$$Q_t = \frac{V}{m}(C_0 - C_t) \tag{1}$$

where $Q_t$ (mg·$g^{-1}$) is the amount of $NH_4^+$–N and $PO_4^{3-}$–P adsorbed by each gram of an adsorbent at time $t$, $C_0$ (mg·$L^{-1}$) denotes the initial concentrations of $NH_4^+$–N and $PO_4^{3-}$–P, $C_t$ (mg·$L^{-1}$) indicates the concentrations of $NH_4^+$–N and $PO_4^{3-}$–P at time $t$, ($C_0 - C_t$) is the change in the concentrations of $NH_4^+$–N and $PO_4^{3-}$–P, $m$ (mg) is the mass of the adsorbent, and $V$ (mL) is the solution volume. The concentrations of $NH_4^+$–N and $PO_4^{3-}$–P remaining in the solution can be calculated once the reaction reaches Eq (2).

$$Q_e = \frac{V}{m}(C_0 - C_e) \tag{2}$$

where $Q_e$ (mg·g$^{-1}$) is the amount of NH$_4^+$–N and PO$_4^{3-}$–P adsorbed by each gram of the adsorbent, and $C_e$ (mg·L$^{-1}$) represents the concentrations of NH$_4^+$–N and PO$_4^{3-}$–P at equilibrium.

To construct adsorption isotherms, adsorption experiments were conducted using mixed adsorption solutions of different concentrations (0, 5, 10, 15, 25, 50, 100, 200, 250 and 400 mg·L$^{-1}$). Adsorption experiments were conducted at 25°C, 35°C, and 45°C, and the pH of the solution was adjusted to 7 using HCl and NaOH. The formula is as stated:

$$Q_e = Q_{max}K_LC_e/(1 + K_LC_e) \tag{3}$$

$$Q_e = K_fC_e^{1/n} \tag{4}$$

where $Q_{max}$ (mg·g$^{-1}$) represents the maximum adsorption capacity of the adsorbent, and $K_f$ [mg·(g·(L·mg$^{-1}$)1/n)$^{-1}$] and $K_L$ (L·mg$^{-1}$) are the constants in the Freundlich and Langmuir models, respectively.

Adsorption kinetics were also investigated. Specifically, 50 mg·L$^{-1}$ PO$_4^{3-}$–P and 100 mg·L$^{-1}$ NH$_4^+$–N solutions were mixed and stirred at 35°C and 150 rpm. The pH of the solution was adjusted to 7. The concentrations of NH$_4^+$–N and PO$_4^{3-}$–P were measured by collecting samples at the designated time points. The adsorption of NH$_4^+$–N and PO$_4^{3-}$–P by the prepared materials was analyzed using the pseudo-first-order, pseudo-second-order, and Weber–Morris models. The corresponding equations are provided below.

$$\ln(Q_e - Q_t) = \ln Q_e - k_1t \tag{5}$$

$$t/Q_e = 1/(k_2Q_e^2) + t/Q_e \tag{6}$$

where $k_1$ (min$^{-1}$) is the adsorption rate constant, $k_2$ (g·mg$^{-1}$·min$^{-1}$) is the adsorption rate constant in the pseudo-second-order model.

$$Q_t = k_{id}t^{1/2} + C \tag{7}$$

where $k_{id}$ is the intra-particle diffusion rate constant (mg/(g·min$^{1/2}$)), $t$ (min) is the adsorption time, and $C$ is a constant that depends on the width of the adsorption boundary layer.

The effect of pH on adsorption was determined at 35°C by changing the pH of the initial solution in the range of 3–11 using either HCl or NaOH. The solid–liquid ratio was maintained at 0.67 g·L$^{-1}$.

Thermodynamic experiments were conducted at 25°C, 35°C, and 45°C to elucidate the natural occurrence and adsorption of NH$_4^+$–N and PO$_4^{3-}$–P. The Gibbs free energy ($\Delta G_0$; kJ·mol$^{-1}$) was calculated based on the experimental data obtained at various temperatures, employing the equation that incorporates changes in enthalpy ($\Delta H_0$; kJ·mol$^{-1}$) and entropy ($\Delta S_0$; kJ·mol$^{-1}$·K$^{-1}$).

$$\Delta G_0 = \Delta H_0 - T\Delta S_0 \tag{8}$$

$$\Delta G_0 = -RT\ln K_C \tag{9}$$

$$K_C = M_W \times 55.5 \times 1000 \times K_L \tag{10}$$

Here, $R$ (8.314 J·mol$^{-1}$·K$^{-1}$) is the gas constant, $K_C$ is the adsorption constant, and $T$ (K) is the absolute temperature. The Langmuir constant, denoted as $K_L$, is the equilibrium constant for adsorption, while $M_W$ (mol) is the quantity of the adsorbent. Using Eqs (8–10), the intercept and slope of the temperature curve of $\Delta G_0$ were determined as $\Delta S_0$, $\Delta H_{0,}$ and $T$.

**Table 1. Basic properties of soil.**

| | Moisture content | pH | CEC | Organic matter | Total phosphorus | Total nitrogen | Available Phosphorus | Hydrolyzable nitrogen |
|---|---|---|---|---|---|---|---|---|
| Primitive soil | % | / | (cmol·kg⁻¹) | (g·kg⁻¹) | (g·kg⁻¹) | (g·kg⁻¹) | (mg·kg⁻¹) | (mg·kg⁻¹) |
| | 11.7 | 7.72 | 18.86 | 18.9 | 0.56 | 0.88 | 24.20 | 49 |

To assess the relative competitiveness of various ions, the adsorption solution was supplemented with anions ($Cl^-$, $HCO_3^-$, $SO_4^{2-}$, and $CO_3^{2-}$) and cations ($Na^+$, $Fe^{3+}$, and $Ca^{2+}$) that are commonly found in wastewater. Their concentrations were varied in increments of 50 $mmol \cdot L^{-1}$ in the range of 0–250 $mmol \cdot L^{-1}$.

$$D = \frac{Q_e - Q_a}{Q_e} \times 100\% \tag{11}$$

## Pot experiments

The potential of MgSBC-0.5(@Al) as a fertilizer was investigated using a pot experiment. The soil was collected from 0–20 cm topsoil at Southeast University (China). The basic properties of soil are shown in Table 1.

MgSBC-0.5(@Al) saturated with $NH_4^+$–N and $PO_4^{3-}$–P (hereafter referred to as MgSBC-0.5(@Al)–NP) was added to the soil at 0.5% of soil dry weight. The leaching process was conducted using the methodology specified by the United States Environmental Protection Agency (USEPA 2014). Mung beans were used as test plants in this experiment. MgSBC-0.5(@Al)–NP, SBC–NP, commercial nitrogen phosphorus fertilizer and SP (0.5 g) were thoroughly mixed into 100 g soil and added to the pot (group A, B, D and E), whereas group C (control) was grown on soil without any additions. Five mung beans were planted in each pot and allowed to grow in a greenhouse at a temperature of around 25°C. Each grid was watered with 2.5 mL of tap water every day. The growth of mung bean sprouts was recorded. On the 14th day, the plants were selected for physical properties determination. The germination rate (GP; %) of mung bean sprouts was calculated using Eq (12).

$$Germination\ percentage(GP) = \frac{n}{N} \times 100\% \tag{12}$$

where $n$ is the number of seeds germinated on the fourteenth day, and $N$ is the total number of seeds.

## Characterization and analytical method

Mo–SS anti-spectrophotometry was employed, in addition to a spectrophotometer (752 N-spectrophotometer; China). The pH of the solution was determined using a pH meter (pHS-3C, China). The $N_2$ adsorption–desorption isotherms were determined using the Quanta-chrome AUTOSORB IQ (United States). The specific surface area ($S_{BET}$) and pore characteristics were measured using the Barrett–Joyner–Halenda method. The morphology and microstructure of the prepared materials were analyzed using scanning electron microscopy (SEM; Gemini 300, Zeiss, Germany). Surface elemental analysis was performed using energy-dispersive X-ray spectroscopy (EDS) at the same locations where SEM was performed. The microstructure and crystal structure were analyzed through X-ray diffraction (XRD; D2 phaser, Bruker, Germany). The composition was analyzed within the 2θ range of 10°–80° captured at the position. The Nickel IS10 catalyst was analyzed through Fourier transform infrared spectroscopy (FTIR) in the wavenumber range of 4000–400 $cm^{-1}$. X-ray photoelectron spectroscopy (XPS; K-Alpha, Thermo Scientific, USA) was performed using an Al K α source

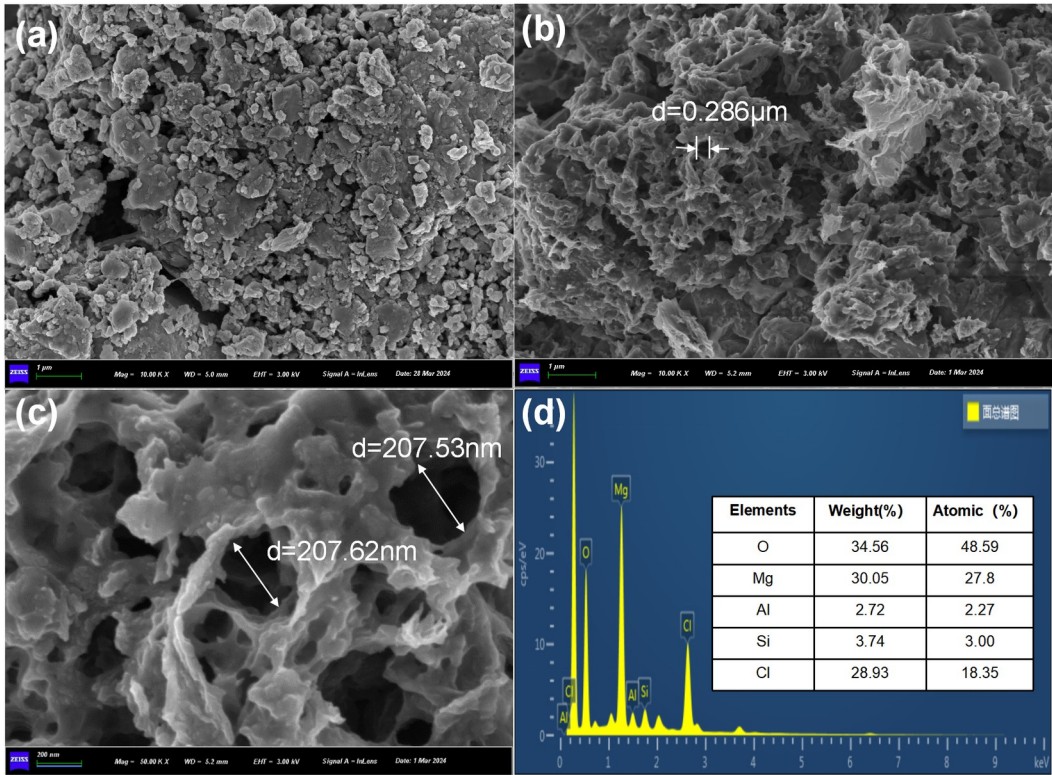

**Fig 1.** (a) SEM images of SBC, SEM images of MgSBC-0.5(@Al) at different expansion ratios (b-c) and (d) EDS spectra of MgSBC-0.5(@Al) corresponding to SEM images.

(hv = 1486.6 eV). Additionally, the pressure within the sample chamber should not exceed $2.0\times$ Perform XPS detection at a pressure of $10^{-7}$ mbar. The elemental composition of the bulk phase of the samples was determined using inductively coupled plasma (ICP)-optical emission spectrometry (OES)/mass spectrometry (MS; Fisher iCAP PRO) and X-ray fluorescence (XRF; SHIMADZU XRF-1800). SPSS statistics 27 was used for significance analysis. Nitrogen and phosphorus contents in mung bean sprouts in five groups (14 days) were determined by Kjeldahl nitrogen analyzer and inductively coupled plasma (ICP)-optical emission spectrometry (OES)/mass spectrometry (MS; Fisher iCAP PRO).

## Results and discussion

### Characterization and morphology

Fig 1 shows the SEM images of the prepared BC specimens, with MgSBC-0.5(@Al) exhibiting a rougher porous structure than SBC (Fig 1A). Additionally, Fig 1B and 1C show that the pore size of MgSBC-0.5(@Al) is greater than 200 nm, providing additional adsorption sites. Fig 1D shows the energy spectrum of MgSBC-0.5(@Al), demonstrating that the sample mainly consists of Mg, Al, Si, Cl, and O [14].

N$_2$ adsorption–desorption isotherms were obtained for SP, SBC, MgSBC-0.5(@Al), and MgSBC-0.5 to determine their $S_{BET}$, total pore volume, and average pore diameter (APD; Table 2). The $S_{BET}$ of the four materials is 4.59, 16.76, 188.86, and 10.44 cm$^2$·g$^{-1}$, respectively. The $S_{BET}$ and total pore size of MgSBC-0.5(@Al) are 11.27 and 5.94 times higher than those of SBC, respectively. The APDs of the four materials are within 2–50 nm, making them

mesoporous materials [15]. In particular, the $S_{BET}$ of MgSBC-0.5 is substantially smaller than that of MgSBC-0.5(@Al), indicating that the electro-assisted modification increased the $S_{BET}$ of BC. However, the APD of MgSBC-0.5(@Al) is low, even though the pore size on the surface of the material is more than 200 nm. Such results were attributed to the blocking of a portion of the pores by the metal oxide or the collapse of the pore structure during high-temperature calcination. The removal of certain substances from the sludge would open some blocked internal pore structures, leading to a reduction of APD. Subsequently, $N_2$ adsorption–desorption isotherms were obtained at 77 K. Based on the IUPAC classification, the $N_2$ adsorption curves of SP, SBC, and MgSBC-0.5(@Al) are all type IV adsorption isotherms, with an $H_3$ hysteresis loop (Fig 2A). At the same time, MgSBC-0.5(@Al) shows no substantial limit adsorption in the $P/P_0$ range of 0.9–1.0. The above indicates that the material contains slit-like pores and has a flake-particle-aggregation pore structure [16,17]. Research found that electro-assisted modification substantially increased the pore size and $S_{BET}$ of BC as well as enriched its pore structure. These findings have important implications for the practical application of BC in adsorption.

The XRD results are shown in Fig 2B, indicating that SP and SBC are mainly composed of $SiO_2$. For MgSBC-0.5(@Al), the diffraction peaks observed at 36.86° (111), 42.83° (200), 62.30° (220), 74.69° (311), and 78.63° (222) were ascribed to MgO. Spinel $MgAl_2O_4$ (PDF # 21–11052), AlOOH (PDF # 21–1307), $Al_2O_3$ (PDF # 10–0425), and $Mg_7Al_4SiO_{15}(OH)_{12}$ (PDF # 47–1866) were also detected. The presence of diffraction peaks that were difficult to identify indicates that the specimens contained small amounts of other composite minerals, such as Mg/Al–Si composite minerals [18,19].

The elemental composition was investigated using XPS. According to the XPS results in Table 3 and Fig 2C, the surface of MgSBC-0.5(@Al) is mainly composed of C, O, N, Si, Al, Ca, and Fe. The contents of Mg and Al in MgSBC-0.5(@Al) are 28.08% and 6.24%, indicating that the oxides of Mg and Al were loaded on the SBC [19]. The loading of $Al_2O_3$ was beneficial for the formation of porous structures in biochar. The presence of $Al^{3+}$ in solution could precipitate with $PO_4^{3-}$–P through strengthened chemical bonds. This might improve the adsorption capacity of MgSBC-0.5(@Al) for $PO_4^{3-}$–P [20]. SP has been reported to contain a small amount of Ca and Fe, which could form amorphous $CaCO_3$ and $FePO_4$) with $PO_4^{3-}$–P, benefiting the adsorption of $PO_4^{3-}$–P [21]. The ICP–MS analysis of the entire sample of MgSBC-0.5(@Al) shows (Table 3) that Mg accounts for 30.3% of the sample, slightly higher than the 28.08% obtained through XPS, indicating that Mg is mainly loaded on the surface of the sample (rather than in the bulk). This might reflect the adsorption of $NH_4^+$–N and $PO_4^{3-}$–P, during which Mg dissolves from the surface of the adsorbent into the solution during exposure or rearranges on the surface after initial dissolution, forming local mineral complexes ($MgHPO_4$). The formation of these complexes can lead to the exposure of carbon lattice, as shown in SEM images (Fig 1B) [22].

Fig 2D shows the FTIR spectra of SP, SBC, and MgSBC-0.5(@Al) in the wavenumber range of 4000–400 $cm^{-1}$. The peak observed at a wavelength of 465 $cm^{-1}$ in SBC was attributed to the Si-O-Si or Al-O-Al vibration. Symmetric and antisymmetric tensile vibrations of Si–O–Si

**Table 2. Comparison of $S_{BET}$, Total pore volume, and APD for SP, SBC, MgSBC-0.5 and MgSBC-0.5(@Al).**

| Parameters | Adsorbent | | | |
|---|---|---|---|---|
| | SP | SBC | MgSBC-0.5(@Al) | MgSBC-0.5 |
| $S_{BET}$ ($m^2 \cdot g^{-1}$) | 4.59 | 16.76 | 188.86 | 10.44 |
| Total pore volume ($cm^3 \cdot g^{-1}$) | 0.031 | 0.037 | 0.19 | 0.044 |
| APD (nm) | 27.37 | 8.86 | 3.95 | 16.98 |

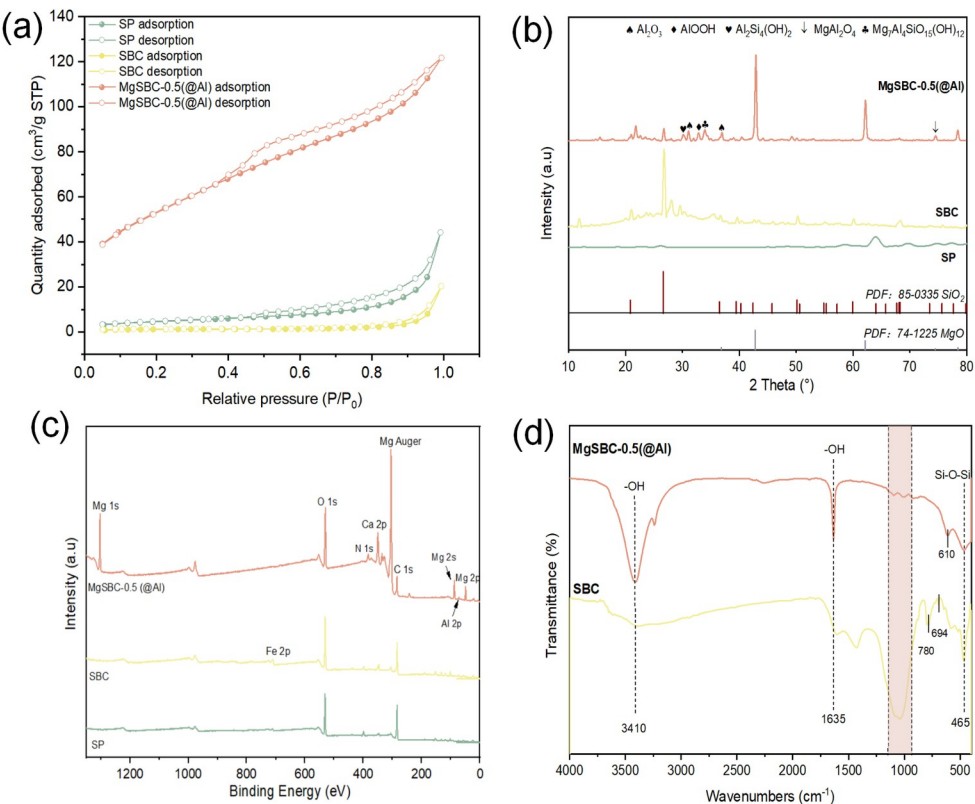

**Fig 2.** (a) Adsorption and desorption curve, (b-d) XRD patterns, XPS spectra and FTIR spectra of SP, SBC and MgSBC-0.5(@Al).

chains are represented by peaks at 694 and 780 cm$^{-1}$, respectively [23,24]. The peaks at 3410 and 1635 cm$^{-1}$ in MgSBC-0.5(@Al) correspond to the stretching vibration of interlayer hydrogen bonding groups and water molecules [25,26]. This indicated that MgSBC-0.5(@Al) may contain bound water in its structure. The peak at 610 cm$^{-1}$ observed in the visible band corresponds to the metal–oxygen bonds (M–O and M–O–M, where M represents either Mg or Al) [27]. Additionally, MgSBC-0.5(@Al) exhibits substantially different absorption peaks than SBC. In region I, a wider peak is observed for SBC, whereas MgSBC-0.5(@Al) shows a number of narrower peaks. This change may have occurred because of the transformation of Si–O–Si bonds into Si–O–Mg and Si–O–Al bonds, which have greater bond lengths and smaller bond angles than the Si–O–Si bonds [28].

## Adsorption isotherm and kinetics

Magnesium acetate solutions with concentrations of 0.1, 0.25, 0.5, and 1 mol·L$^{-1}$ were prepared. The adsorption capabilities of MgSBC-0.1(@Al), MgSBC-0.25(@Al), MgSBC-0.5(@Al),

**Table 3. The elemental composition (atomic%) measured by XPS and ICP-MS.**

| Analysis methods | Sample | C | O | Si | Mg | Al | Ca | Fe | P | N | Cl |
|---|---|---|---|---|---|---|---|---|---|---|---|
| XPS | SP | 61.05 | 30.24 | 4.44 | 0.52 | / | 1.34 | 1.06 | 1.03 | 0.33 | / |
| | SBC | 47.70 | 36.14 | 7.44 | 0.91 | / | 2.97 | 2.18 | 1.83 | 0.83 | / |
| | MgSBC-0.5(@Al) | 14.34 | 23.69 | 0.43 | 28.08 | 6.14 | 4.56 | 9.23 | 2.01 | 0.49 | 11.03 |
| ICP-MS | MgSBC-0.5(@Al) | / | / | / | 30.3 | 6.3 | / | / | / | / | |

and MgSBC-1(@Al)) for $NH_4^+$–N and $PO_4^{3-}$–P were compared using S1A Fig MgSBC-0.5 (@Al) shows the highest adsorption capacities for $NH_4^+$–N and $PO_4^{3-}$–P at 308 K, 28.22 and 58.73 mg·g$^{-1}$, respectively. The four materials were characterized by XRD. It can be seen from S1B Fig that the peak of MgO in MgSBC-0.5(@Al) is higher than that of the other three materials, indicating that MgSBC-0.5(@Al) is loaded with more MgO. $Mg^{2+}$ might play a role in the formation of struvite crystals. This transformation was not beneficial for the adsorption of $NH_4^+$–N and had minimal impact on the adsorption of $PO_4^{3-}$–P. Therefore, setting a limit for Mg-loading was essential [29].

The adsorption equilibrium isotherms of MgSBC-0.5(@Al) for $NH_4^+$–N and $PO_4^{3-}$–P were obtained at 298, 308, and 318 K. The adsorption process of $NH_4^+$–N may be accurately described by the Langmuir and Freundlich models (Fig 3A and 3B) (Table 4) (Eqs (3) and (4)). These results indicate that MgSBC-0.5(@Al) adsorbs $NH_4^+$–N through both monolayer and multilayer adsorption. The adsorbate was evenly distributed on the surface of the adsorbent, resulting in saturation [30]. The adsorption of $PO_4^{3-}$–P on MgSBC-0.5(@Al) at 298 K can be better described by the Langmuir model, as shown in Fig 3C and 3D. This suggests that the adsorption of $PO_4^{3-}$–P was likely monolayer. The Langmuir equation was used to determine the maximum adsorption capacities of MgSBC-0.5(@Al) for $NH_4^+$–N and $PO_4^{3-}$–P, yielding 65.19 and 92.10 mg·g$^{-1}$, respectively, at 298 K. The maximum adsorption capacities of SBC for $NH_4^+$–N and $PO_4^{3-}$–P are 11.01 and 14.66 mg·g$^{-1}$ (S2A and S2B Fig). At the same time, the $S_{BET}$ of MgSBC-0.5(@Al) is 18.09 times larger than that of SBC. The higher adsorption capacity of MgSBC-0.5(@Al) could not be attributed solely to the chemical adsorption on MgO, AlOOH, $Al_2O_3$, and other Mg/Al/Si composites on the BC surface. Physical adsorption and other factors also likely contributed to it [31].

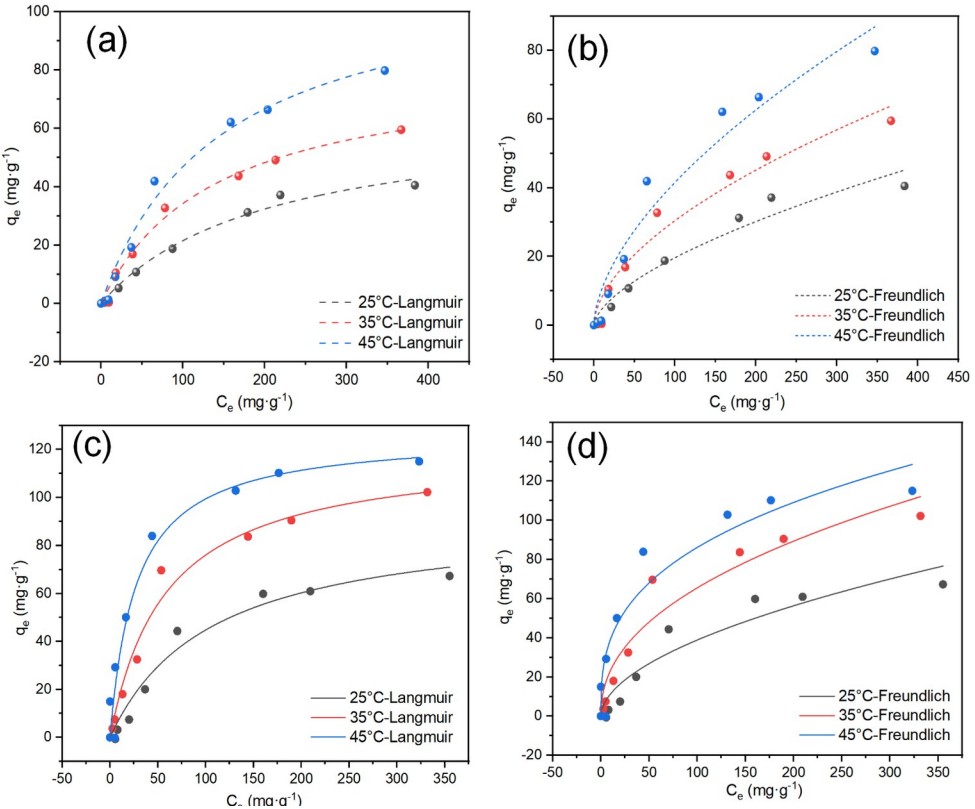

**Fig 3.** (a, b) $NH_4^+$–N and (c, d) $PO_4^{3-}$–P adsorption isotherms of MgSBC-0.5(@Al) at different temperatures.

**Table 4. Isotherm adsorption model parameters of MgSBC-0.5(@Al) and SBC at 298K.**

| Isotherm models | Parameters | Adsorbent-adsorbate | | | |
|---|---|---|---|---|---|
| | | MgSBC-0.5(@Al)-a | MgSBC-0.5(@Al)-b | SBC-a | SBC-b |
| Langmuir | $Q$ (mg·g$^{-1}$) | 65.19 | 92.10 | 11.01 | 14.66 |
| | $K_L$ (L·mg$^{-1}$) | 0.0049 | 0.0095 | 0.024 | 0.011 |
| | $R^2$ | 0.987 | 0.967 | 0.925 | 0.992 |
| Freundlich | $K_F$ | 1.13 | 3.30 | 1.10 | 0.84 |
| | 1/n | 0.62 | 0.54 | 0.39 | 0.46 |
| | $R^2$ | 0.954 | 0.905 | 0.919 | 0.988 |

Notes: ("a": the result of $NH_4^+$-N, and "b": the result of $PO_4^{3-}$-P).

$NH_4^+$–N and $PO_4^{3-}$–P adsorption capabilities of MgSBC-0.5(@Al) might be enhanced by increasing the reaction temperature. As presented in Table 5, with the increase in temperature from 298 to 318 K, the maximum adsorption capacities of $NH_4^+$–N and $PO_4^{3-}$–P increase from 65.19 to 115.28 mg·g$^{-1}$, and from 92.10 to 126.49 mg·g$^{-1}$, respectively.

The values of $\Delta H$ and $\Delta G$ were obtained from S2C Fig, indicating that the adsorption process is endothermic ($\Delta H > 0$; Table 6) [32]. The value of $\Delta G$ is negative and increases in absolute value with increasing temperature. Thus, the adsorption occurred spontaneously, and an increase in temperature enhanced its efficiency. The $\Delta S > 0$ indicates that disorder and degrees of freedom at the solid–liquid interface increase during adsorption. This increase in disorder favored the occurrence of chemical adsorption [33].

The adsorption of $NH_4^+$–N and $PO_4^{3-}$–P on MgSBC-0.5(@Al) was studied using pseudo-first-order and pseudo-second-order models. Fig 4A and 4B show the adsorption of $NH_4^+$–N and $PO_4^{3-}$–P on MgSBC-0.5(@Al) reaches equilibrium at around 90 and 270 min, respectively. Fitted with Eqs (5) and (6), the adsorption capacities of MgSBC-0.5(@Al) for $NH_4^+$–N and $PO_4^{3-}$–P are 33.40 and 67.21 mg·g$^{-1}$, respectively. Table 7 shows that the R$^2$ of the pseudo-second-order kinetic model (0.988) is higher than that of the pseudo-first-order kinetic model (0.973) for $PO_4^{3-}$–P adsorption. This implies that $PO_4^{3-}$–P adsorption was primarily a chemical process. This mechanism involved multiple components, including the valence force, the

**Table 5. Isotherm adsorption model parameters of MgSBC-0.5(@Al) at 298, 308, and 318 K.**

| Models | Parameters | Values | | |
|---|---|---|---|---|
| | | 298 K | 308 K | 318 K |
| Langmuir-a | $Q$ (mg·g$^{-1}$) | 65.19 | 81.79 | 115.28 |
| | $K_L$ (L·mg$^{-1}$) | 0.0049 | 0.0072 | 0.0069 |
| | $R^2$ | 0.987 | 0.988 | 0.984 |
| Freundlich-a | $K_F$ | 1.13 | 2.20 | 2.71 |
| | 1/n | 0.62 | 0.57 | 0.50 |
| | $R^2$ | 0.954 | 0.958 | 0.948 |
| Langmuir-b | $Q$ (mg·g$^{-1}$) | 92.10 | 120.12 | 126.49 |
| | $K_L$ (L·mg$^{-1}$) | 0.0095 | 0.017 | 0.037 |
| | $R^2$ | 0.967 | 0.981 | 0.954 |
| Freundlich-b | $K_F$ | 3.30 | 8.36 | 17.91 |
| | 1/n | 0.54 | 0.45 | 0.34 |
| | $R^2$ | 0.905 | 0.930 | 0.889 |

Notes: ("a": is the result of $NH_4^+$-N, and "b" is the result of $PO_4^{3-}$-P).

**Table 6. Calculated thermodynamic parameters.**

| Pollutant | Temperature (k) | $K_L$ (L·mg$^{-1}$) | $K_C$ | $\Delta H$ (kJ·mol$^{-1}$) | $\Delta S$ (kJ·mol$^{-1}$·K$^{-1}$) | $\Delta G$ (kJ·mol$^{-1}$) |
|---|---|---|---|---|---|---|
| NH$_4^+$-N | 298 | 0.0049 | 4929.51 | 12.73 | 0.11 | -21.07 |
| | 308 | 0.00720 | 7189.28 | | | -22.74 |
| | 318 | 0.0069 | 6849.32 | | | -23.35 |
| PO$_4^{3-}$-P | 298 | 0.0095 | 49907.02 | 52.60 | 0.27 | -26.80 |
| | 308 | 0.017 | 90845.55 | | | -29.24 |
| | 318 | 0.037 | 193772.20 | | | -32.19 |

exchange or sharing of electrons between the adsorbent and the adsorbate, and the potential synthesis of new compounds. At the same time, the $R^2$ of the pseudo-first-order kinetic model (0.988) is higher than that of the pseudo-second-order kinetic model (0.986) for the adsorption of NH$_4^+$–N. This suggests that NH$_4^+$–N was adsorbed primarily through physical adsorption [34]. However, the diffusion mechanism of the adsorbent could not be accurately identified based on the pseudo-first-order and pseudo-second-order models. Therefore, the Weber–Morris equation was employed to comprehensively investigate the rate-limiting mechanism of the adsorption process [35].

If adsorption was constrained solely by intra-particle diffusion, the regression line would pass through the origin. However, the intercept (*C*) of the NH$_4^+$–N and PO$_4^{3-}$–P adsorption

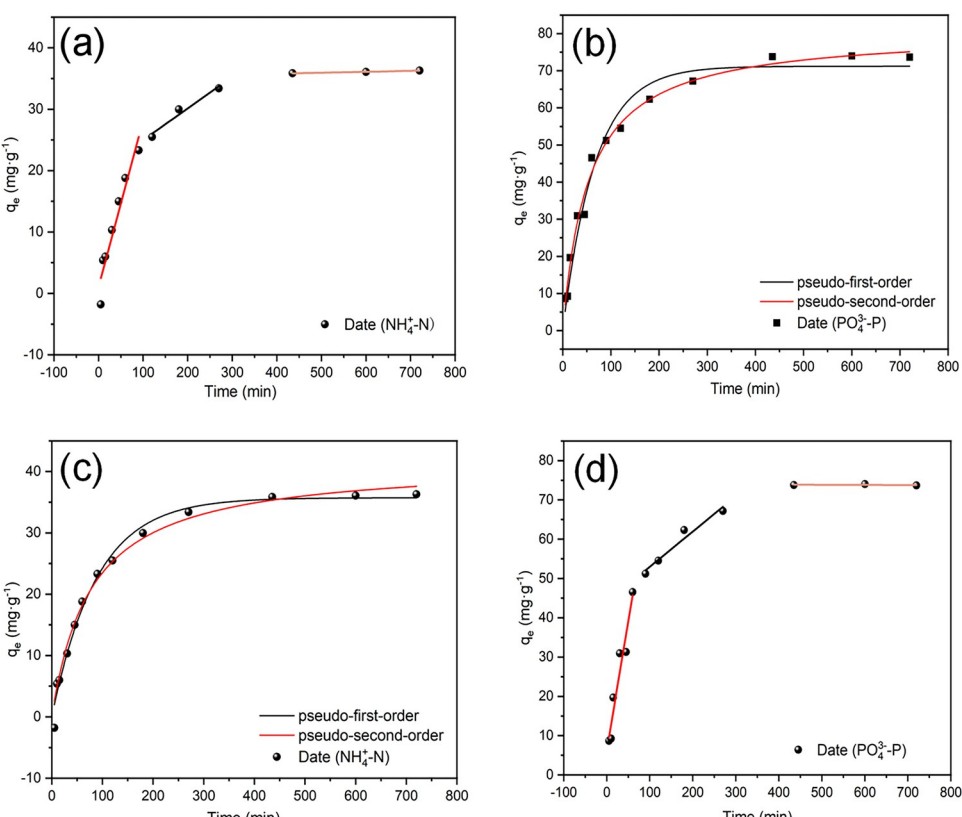

**Fig 4.** (a-b) The adsorption kinetics of MgSBC-0.5(@Al) for ammonia nitrogen: pseudo first-order and pseudo second-order models and Weber Morris models, and (c-d) pseudo first-order, pseudo second-order models and Weber Morris models for phosphate.

**Table 7. Adsorption kinetics parameters of MgSBC-0.5(@Al) adsorption of NH4+-N and PO43—P.**

| Isotherm models | Parameters | Adsorbent-adsorbate | |
| --- | --- | --- | --- |
| | | MgSBC-0.5(@Al)- $NH_4^+$-N | MgSBC-0.5(@Al)- $PO_4^{3-}$—P |
| Pseudo-first-order | $Q_e$ (mg·g$^{-1}$) | 35.73 | 71.21 |
| | $k_1$ (min$^{-1}$) | 0.01137 | 0.015 |
| | $R^2$ | 0.988 | 0.973 |
| Pseudo-second-order | $Q_e$ (mg·g$^{-1}$) | 41.60 | 80.68 |
| | $k_2$ (g (mg·min)$^{-1}$) | $3.1175912\times10^{-4}$ | $2.3189232\times10^{-4}$ |
| | $R^2$ | 0.98578986 | 0.98801 |
| Weber-Morris | $k_{i2}$ (mg g$^{-1}$ min$^{-0.5}$) | 0.05152 | 0.08979 |
| | $R^2$ | 0.92805 | 0.93755938 |

curve is not equal to zero (Fig 4C and 4D) (Eq (7)), suggesting that intra-particle diffusion is not the sole governing force of adsorption [36]. The adsorption of $NH_4^+$–N and $PO_4^{3-}$–P exhibits a multilinear pattern. The first curve in the graph shows the fast adsorption of contaminants over numerous binding sites within a brief timeframe, accompanied by cation exchange for $NH_4^+$–N. The second curve indicates that diffusion within the particles played a major role, and that $NH_4^+$–N and $PO_4^{3-}$–P diffused into the particles. The last curve exhibits a decrease in the rate of diffusion within the particles because of the lower pollutant concentrations in the solution. The second-stage curves for $NH_4^+$–N and $PO_4^{3-}$–P exhibit higher $R^2$ values (0.99 and 0.96, respectively). The Weber–Morris model was used to explain the adsorption of $NH_4^+$–N and $PO_4^{3-}$–P on BC. This model suggests that diffusion within the particles played a major role in adsorption. However, other mechanisms also limited the rate of adsorption. The adsorption of $NH_4^+$–N and $PO_4^{3-}$–P was likely controlled by several factors, such as rapid adsorption on the outer surface, ion exchange between Mg/Al, and simultaneous diffusion inside the SBC [37,38].

## Effects of different factors on $NH_4^+$–N and $PO_4^{3-}$–P capture

With the increase in pH from 3 to 7, the adsorption capacity for $NH_4^+$–N gradually increases, reaching a maximum of 38.48 mg·g$^{-1}$ (Fig 5A). With the increase in pH from 7 to 9, no changes are observed. At the same time, at pH > 9, the adsorption capacity for $NH_4^+$–N decreases. For $PO_4^{3-}$–P, the adsorption capacity gradually increases with pH, from 3 to 7,

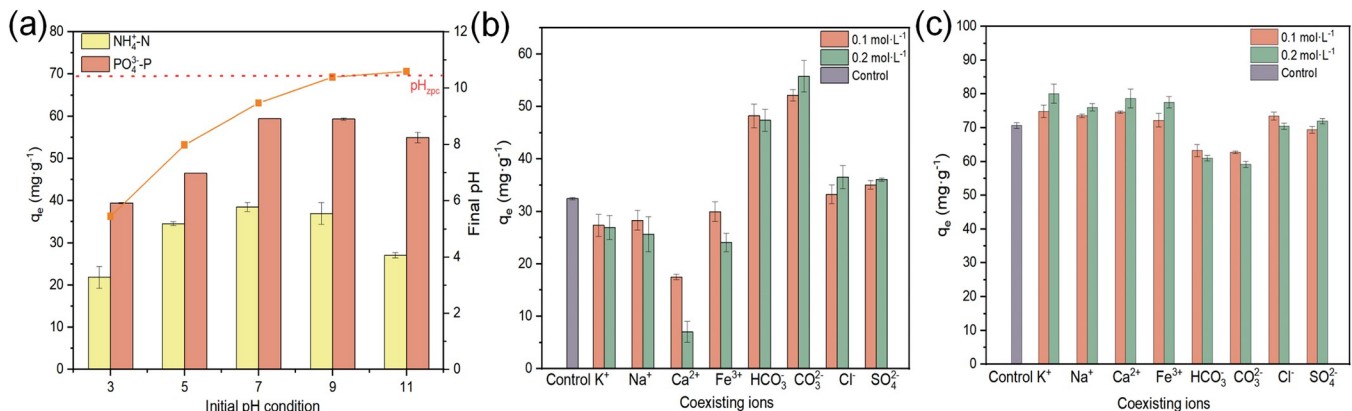

**Fig 5.** (a) Effect of initial pH of solution on adsorption of ammonia nitrogen and phosphorus by MgSBC-0.5(@Al), (b-c) The effect of coexisting ions on the removal of ammonia nitrogen and phosphorus. (t = 35°C, pH = 7, adsorbent dosage 0.67 g·L$^{-1}$).

reaching 59.44 mg·g$^{-1}$. At pH > 7, the adsorption capacity for $PO_4^{3-}$–P decreases. At the same time, at pH < 3, the adsorption capacity of MgSBC-0.5(@Al) for $NH_4^+$–N and $PO_4^{3-}$–P significantly decreases. At pH > 9, the adsorption capacities for $NH_4^+$–N and $PO_4^{3-}$–P also notably decrease. Such results were attributed to (1) the reduction of dissociated carboxyl groups in BC at low pH [39], causing a relatively low adsorption capacity for $PO_4^{3-}$–P. (2) $PO_4^{3-}$–P existed in different forms at different pH: $H_2PO_4^-$ ($pk_1$ = 2.15) at pH < 7.20, $HPO_4^{2-}$ ($pk_2$ = 7.20) at pH of 7.20–12.33, and $PO_4^{3-}$ at pH > 12.33. The lower adsorption free energy of $H_2PO_4^-$ compared to $HPO_4^{2-}$ indicates that the former would be more readily adsorbed by the adsorbent. Consequently, the adsorption of $PO_4^{3-}$–P was strongly affected by pH in the range of 7.20–12.22 [40,41]. (3) The zero point potential (pH$_{ZPC}$) of MgSBC-0.5(@Al) was 10.45, indicating the presence of hydroxyl ligands on its surface, which facilitated the adsorption of $PO_4^{3-}$–P. When the initial pH>pHzpc, the surface of MgSBC-0.5(@Al) carried a negative charge. There was a repulsive force between $PO_4^{3-}$–P and MgSBC-0.5(@Al), which inhibited electrostatic adsorption. However, in this case, chemical interactions such as precipitation reactions between $Mg^{2+}$and $HPO_4^{2-}$ might contribute to the removal of $PO_4^{3-}$–P [42,43]. (4) Electrostatic repulsion between negatively charged $PO_4^{3-}$–P anions and the deprotonated surface of BC strengthened with the increase in the concentration of $OH^-$ in the solution at pH of 9–11. When the pH rose to about 11, a high pH corresponded to an increase in the concentration of hydroxide ions ($OH^-$) in the solution. $NH_4^+$-N was mainly in the form of $NH_3·H_2O$. Therefore, the adsorption capacity of MgSBC-0.5(@Al) for $NH_4^+$-N was weakened. So the overall adsorption effect was decreased. [29,44]. At equilibrium, the pH is maintained at 8–10, which was in the pH zone conducive to struvite crystallization (pH > 8.5). This indicated that the modification of SBC considerably increased its adsorption for $NH_4^+$–N and $PO_4^{3-}$–P.

The effect of co-presence of several cations and anions, such as $K^+$, $Ca^{2+}$, $Fe_3^+$, $Na^+$, $SO_4^{2-}$, $Cl^-$, $HCO_3^-$, and $CO_3^{2-}$, in wastewater on the removal of $NH_4^+$–N and $PO_4^{3-}$–P by MgSBC-0.5(@Al) was investigated. The results of the interference test for single coexisting ions are shown in Fig 5B and 5c. $K^+$ and $Na^+$ in the solution competed with $NH_4^+$–N and reacted with $PO_4^{3-}$–P to form $KMgPO_4·7H_2O$ and $NaMgPO_4·7H_2O$. $Ca^{2+}$ and $Fe^{3+}$ in the solution would immediately react with $PO_4^{3-}$–P to form amorphous $CaCO_3$ and $FePO_4$, respectively, producing irregular crystals and seriously hindering the crystallization of struvite. Hence, the presence of $K^+$, $Na^+$, $Ca^{2+}$, and $Fe^{3+}$ in the solution led to an increase in $PO_4^{3-}$–P adsorption capacity and a decrease in $NH_4^+$–N adsorption capacity [45–47]. $SO_4^{2-}$ and $Cl^-$ show negligible effects on adsorption. However, the presence of $HCO_3^-$ or $CO_3^{2-}$ promoted the adsorption of $NH_4^+$–N by MgSBC-0.5(@Al). One potential explanation is that the presence of these two ions might have resulted in a pH increase, strengthening electrostatic attraction between $NH_4^+$–N and the adsorbent, thus enhancing the adsorption efficiency. At the same time, the adsorption capacity for $PO_4^{3-}$–P decreased. This likely occurred because carbonate inhibited the precipitation of struvite, resulting in a pH increase and a consequent increase in the electrostatic repulsion between $PO_4^{3-}$–P and the adsorbent [48].

To study the N and P removal performance of MgSBC-0.5(@Al) on real samples, adsorption experiments were conducted using anaerobic-fermentation biogas slurry. The total phosphorus and total nitrogen concentrations in the biogas slurry were 31.4 ± 1.0 and 50.1 mg·L$^{-1}$, respectively, and its pH was 7.7 ± 0.2. As shown in S3 Fig, with the increase in solid–liquid ratio from 0.33 to 6.70 g·L$^{-1}$, the removal rates of total phosphorus and total nitrogen increase. The removal rates reach 71.34% and 69.69%, respectively, and then stabilize. At the same time, SBC at a dosage of 6.7 g·L$^{-1}$ shows the total nitrogen and total phosphorus removal rates of only 10.06% and 1.03%, respectively. This indicated that for MgSBC-0.5(@Al), 6.70 g·L$^{-1}$ was the optimal solid–liquid ratio for treating actual biogas slurry, and modified sludge-based biochar had a better treatment effect on actual biogas slurry. But compared to the simulated

solution, the adsorbent had a poor effect on nitrogen and phosphorus in the biogas slurry. This was attributed to the presence of substances in the biogas slurry competing with nitrogen and phosphorus for adsorption sites on MgSBC-0.5(@Al), resulting in decreased adsorption capacity.

## Mechanism of $NH_4^+$–N and $PO_4^{3-}$–P capture

To validate the adsorption mechanism, further characterization was conducted through FTIR and XPS. Fig 6A shows the FTIR spectrum. The bound water–OH vibration peak of MgSBC-0.5(@Al) shifts from 3410 to 3393 cm$^{-1}$ after adsorption and then to 3470 cm$^{-1}$ after desorption (Fig 6A). This transition suggests that adsorption–desorption involves the formation and breakage of hydrogen bonds and electrostatic attraction between pollutants and the adsorbent [49]. The peaks at 586 and 610 cm$^{-1}$ remained within the range of metal–oxygen bonding (400–650 cm$^{-1}$). In particular, MgSBC-0.5(@Al)–NP shows a peak near the 847 cm$^{-1}$, which was attributed to the deformation of–OH linked to $Mg^{2+}$ produced by $Mg(OH)_2$. The observed shift in the peak around the wavelength 1636 cm$^{-1}$ was correspond to the adsorption of $NH_4^+$–N, resulting in a widening of the peak. Additionally, the–OH vibration peak transformed into a symmetric bending vibration peak of N–H in $NH_4$ units after adsorption. Furthermore, the peak at 1082 cm$^{-1}$ was assigned to the asymmetric stretching vibration of P–O. Considering crystal structure, struvite consists of three distinct functional groups: (1) $PO_4$ tetrahedron, (2) Mg·6H$_2$O octahedron, and (3) $NH_4$ groups. These groups are connected by hydrogen bonds. The presence of $NH_4^+$–N and $PO_4^{3-}$–P adsorbed on the BC was confirmed through FTIR [50].

The XRD images of MgSBC-0.5(@Al) before and after $NH_4^+$-N and $PO_4^{3-}$P adsorption are compared to further elucidated the adsorption mechanism (Fig 6B). It could see that the MgO peak in the XRD spectrum of biochar is weakened. A series of XRD peaks appeared at 20.85˚ (111), 21.45˚ (021), 25.72˚ (200) and 30.19˚ (012) for MgSBC-0.5(@Al), corresponding to the $MgNH_4PO_4 \cdot H_2O$ diffraction peak. This indicated that the synthesis of struvite crystallization.

XPS was used to analyze the changes in the chemical composition of MgSBC-0.5(@Al) caused by adsorption. Fig 6C shows that the N 1s and P 2p characteristic peaks of MgSBC-0.5(@Al)–NP are notably more prominent than those of MgSBC-0.5(@Al) and Des-MgSBC-0.5(@Al)–NP. This suggests that $NH_4^+$–N and $PO_4^{3-}$–P were adsorbed. Additionally, the contents of trace elements (Mg, Ca, Al, and Fe) decreased after adsorption, as indicated in Table 8 [51]. As shown in Fig 7A, the Mg 1s peak shifted from 1303.48 to 1304.08 eV after adsorption, indicating a change in the chemical state of Mg. The O 1s peak is composed of Mg–O (531.2 and 529.6 eV) and Al(OH)$_3$ (or Mg/Al composite; 531.10 eV) peaks (Fig 7B) [52,53]. After the adsorption of $NH_4^+$–N, a weak peak corresponding to the binding energy of N 1s at 400.01 eV

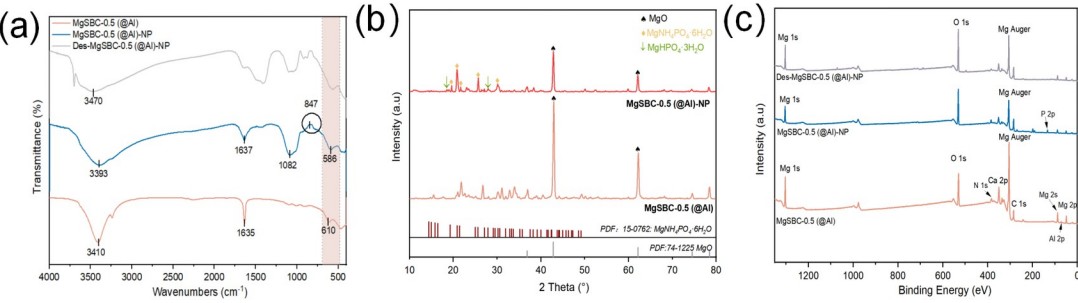

**Fig 6.** (a-c) FTIR, XRD and XPS spectra of MgSBC-0.5(@Al) adsorption of $NH_4^+$—N and $PO_4^{3-}$—P.

**Table 8. Element composition (atomic%) of MgSBC-0.5(@Al) and MgSBC-0.5(@Al)-NP measured by XPS and XRF.**

| Analysis methods | Sample | C | O | Si | Mg | Al | Ca | Fe | P | N | Other elements |
|---|---|---|---|---|---|---|---|---|---|---|---|
| XPS | MgSBC-0.5(@Al) | 14.34 | 23.69 | 0.43 | 28.08 | 6.14 | 4.56 | 9.23 | 2.01 | 0.49 | / |
| | MgSBC-0.5(@Al)-NP | 8.97 | 24.73 | 1.68 | 20.57 | 3.64 | 1.36 | 6.05 | 26.91 | 5.09 | / |
| XRF | MgSBC-0.5(@Al)-NP | 6.90 | 27.26 | 7.14 | 16.18 | 3.97 | 0.93 | 7.62 | 14.65 | 10.32 | 5.03 |

appears in the XPS spectrum, proving that $NH_4^+$ ions were successfully adsorbed on the BC (Fig 7C). The P 2p peaks at 132.9 and 133.75 eV correspond to $HPO_4^{3-}$ and $PO_4^{3-}$, respectively (Fig 7D) [54–56]. Although the solubility of MgO and $Al_2O_3$ is very low, the introduction of $PO_4^{3-}$–P anion into the solution promoted their dissolution, resulting in the formation of greater amounts of insoluble salts ($MgHPO_4$ and $MgNH_4PO_4 \cdot 6H_2O$). Therefore, the adsorption on MgSBC-0.5(@Al) likely involved precipitation reactions.

The XRF spectrum of the entire MgSBC-0.5(@Al)–NP sample (Table 9) shows the presence of insoluble mineral phases on the surface and in the bulk of MgSBC-0.5(@Al). The most abundant inorganic component of MgSBC-0.5(@Al)–NP was Mg; additionally, the sample contains N (10.32%) and P (14.65%), and the N content was considerably higher than that detected using XPS. It might be that only a part of the $NH_4^+$–N was adsorbed on the material surface, and the other part physically adsorbed in the pores of the biochar, which was consistent with the conclusion of the kinetic model [57]. The removal of $NH_4^+$–N was mainly controlled by physical adsorption. The Al, Ca, and Fe contents of MgSBC-0.5(@Al)–NP are consistent with the XPS results, indicating that most of these metal oxides were loaded on the

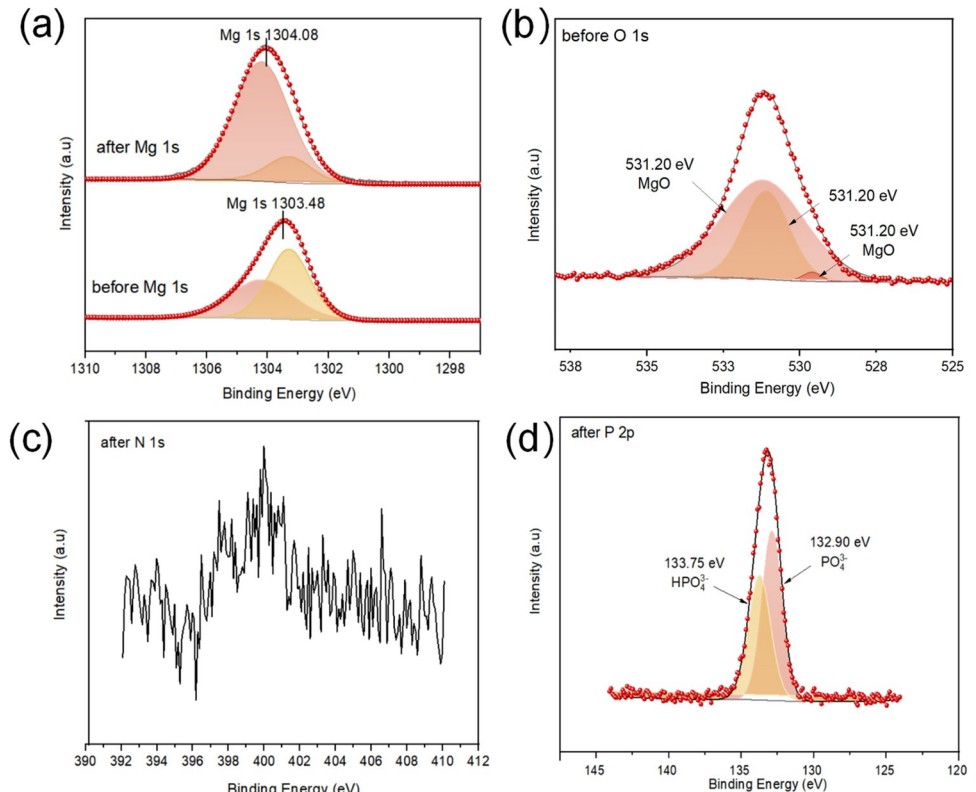

**Fig 7.** (a) XPS spectra of Mg 1s, (b) XPS spectra of O 1s before utilization, and (c-d) XPS spectra of N 1s and P 2p after utilization.

**Table 9. Effects of five groups on yield and characters of mung bean sprouts.**

| Group | Stem Length (cm) | Root Length (cm) | Average wet weight (g) | Average dry weight (g) | Nitrogen (g·kg$^{-1}$) | Phosphorus (g·kg$^{-1}$) |
|---|---|---|---|---|---|---|
| A | 12.58 ± 3.33b | 9.15 ± 4.10ab | 0.42 ± 0.041b | 0.085 ± 0.0068b | 41.28 | 3.86 |
| B | 11.52 ± 2.52c | 8.21 ± 3.17ab | 0.36 ± 0.029c | 0.069 ± 0.0089c | 30.07 | 2.21 |
| C | 8.92 ± 3.89c | 7.02 ± 2.54b | 0.32 ± 0.024d | 0.066 ± 0.0038cd | 29.12 | 2.16 |
| D | 16.79 ± 1.41c | 9.34 ± 2.32a | 0.65 ± 0.035a | 0.092 ± 0.0080a | 57.17 | 5.96 |
| E | 8.58 ± 0.92a | 7.99 ± 1.38 ab | 0.34 ± 0.033c | 0.063 ± 0.0048d | 29.66 | 2.17 |

Note: 1: Repeat 4 times for each group of samples; 2: Values with superscript letters a, b, c and d are significantly different across columns (p<005).

surface of the material. Silicon accounts for 7.14%, indicating that Si was distributed inside the material. The overall spectrum is shown in S4 Fig.

The adsorption mechanisms were inferred based on the physical and chemical properties of MgSBC-0.5(@Al), $S_{BET}$, XRD, FTIR, XPS, and other relevant data. The adsorption mechanism was concluded to include the following phenomena. (1) The physical adsorption capacity of biochar was enhanced through electro-assisted modification with Mg and thermal modification, resulting in a considerable increase in the surface area of MgSBC-0.5(@Al). These processes also increased the number of adsorption sites on the material, improving its physical adsorption capacity. (2) The reduction of MgO and $Al_2O_3$ to $Mg(OH)_2$ and $Al(OH)_3$ at the solid–liquid interface caused an increase in pH. At pH of the aqueous solution above 2.15, $H_2PO_4^-$, $HPO_4^{2-}$, and $PO_4^{3-}$ were the main forms of $PO_4^{3-}$–P. Pollutants were removed by electrostatic adsorption [58]. (3) Struvite crystallization was the primary mechanism of the removal of $NH_4^+$–N and $PO_4^{3-}$–P. At the solid–liquid interface, MgO formed a hydrated intermediate ($Mg(OH)_2$), which reacted with $NH_4^+$–N and $PO_4^{3-}$–P in the solution to form $MgNH_4PO_4·6H_2O$. The main reactions are shown in Eqs (12–18).

$$MgO + H_2O \rightarrow Mg^{2+} + 2OH^- \tag{13}$$

$$MgO + H_2O \rightarrow MgOH^+ + OH^- \tag{14}$$

$$MgOH^+ + HPO_4^{2-} \rightarrow MgOH^+ \cdots HPO_4^{2-} \tag{15}$$

$$Mg^{2+} + HPO_4^{2-} \rightarrow MgHPO_4 \tag{16}$$

$$Mg^{2+} + NH_4^+ + H_2PO_4^- + 6H_2O = MgNH_4PO_4 \cdot 6H_2O \downarrow + 2H^+ \tag{17}$$

$$Mg^{2+} + NH_4^+ + HPO_4^{2-} + 6H_2O = MgNH_4PO_4 \cdot 6H_2O \downarrow + H^+ \tag{18}$$

$$Mg^{2+} + NH_4^+ + PO_4^{3-} + 6H_2O = MgNH_4PO_4 \cdot 6H_2O \downarrow \tag{19}$$

## Nitrogen and phosphorus recovery

Pot experiments were employed to test the applicability of MgSBC-0.5(@Al)–NP as an N and P fertilizer for plant growth. The germination rates of mung bean seeds in the five groups were 95%, 90%, 95%,90% and 85%, respectively, indicating that the addition of BC had little effect on germination rate. The photos of the plants at days 1, and 14 are shown in Fig 8A and 8B.

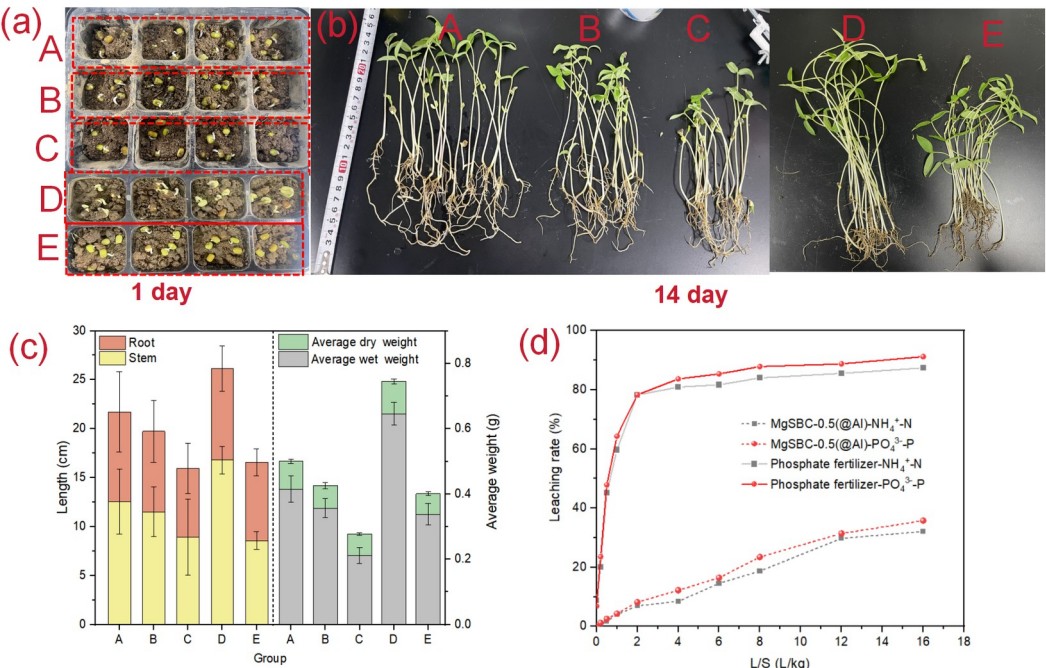

**Fig 8.** (a-b) Growth of mung beans on days 1 and 14, (c) Root length, stem length, average wet weight and average dry weight of five groups of mung beans on day 14, and (d) $NH_4^+$-N and $PO_4^{3-}$-P leaching rates of MgSBC-0.5(@Al)-NP and commercial fertilizers.

However, as shown in Fig 8C, on day 14, mung bean sprouts in group A are on average substantially larger than those of group B, group C and group (Fig 8C). But it was lower than that group D. The average wet weight and dry weight of seedlings in group A were 0.42 g and 0,088 g (Table 9). After calculation, for stem length, average wet weight and average dry weight, there were significantly differences between groups A and E. And Group A was significantly different from the other three groups (Groups B, C, and D) ($p < 005$). But for Root length. There was only a significantly difference between groups C and D. Although the growth and development of bean sprouts in group D (with the addition of commercial fertilizer) was slightly higher than that of group A, but the growth and development of mung bean sprouts in group A was much better than that of groups B, C and E. The contents of nitrogen and phosphorus in mung bean sprouts in group A (41.28 mg·kg$^{-1}$ and 3.86 mg·kg$^{-1}$) were also higher than those in groups B, C and D. The growth and development of mung bean sprouts in group A was promoted by additional N and P in the soil. Notably, Mg-loaded BC had also been reported to promote crop growth [59], in part because its stable carbon structure improved soil microbial activity [60]. Therefore, MgSBC-0.5(@Al)–NP could be used as an N and P fertilizer.

To investigate the sustained release effect of MgSBC-0.5(@Al)–NP on $NH_4^+$–N and $PO_4^{3-}$–P after adsorption, leaching experiments were conducted. The leaching results of $NH_4^+$–N and $PO_4^{3-}$–P are shown in Fig 8D. The leaching characteristics of MgSBC-0.5(@Al)–NP were different from those of commercial fertilizers. The release of $NH_4^+$–N and $PO_4^{3-}$–P in MgSBC-0.5(@Al)-NP showed a gradually increasing trend through the leaching process. When the L/S ratio reached 2.0 L·kg$^{-1}$, the cumulative leaching amounts of $NH_4^+$–N and $PO_4^{3-}$–P were only 2.53 mg·g$^{-1}$(6.98%) and 6.05 mg/g (8.21%). At the end of the experiment, the cumulative leaching amounts of $NH_4^+$–N and $PO_4^{3-}$–P were 11.54 mg·g$^{-1}$(32.12%) and 26.36 mg·g$^{-1}$ (35.78%), which were 2.72 times and 2.54 times lower than those of commercial fertilizers, respectively.

The generated $MgHPO_4 \cdot 3H_2O$ and $MgNH_4PO_4 \cdot 6H_2O$ reduced the leaching of $NH_4^+$–N and $PO_4^{3-}$–P, forming a sustained release effect of MgSBC-0.5(@Al)-NP [61].

## Conclusions

Mg/Al-modified SBC was prepared via the electro-assisted modification of sludge followed by pyrolysis. The specific surface area of MgSBC-0.5(@Al) was 11.27 times higher than that of SBC and 18.06 times higher than that of MgSBC-0.5. The surface of biochar was covered with oxygen-containing metal complexes, including MgO, AlOOH, and $Al_2O_3$. The maximum adsorption capacities of MgSBC-0.5(@Al) for $NH_4^+$–N and $PO_4^{3-}$–P were 65.19 and 92.10 $mg \cdot g^{-1}$, higher by 4.45 and 6.28 times than those of SBC, respectively. The adsorption experiments and material characterization results confirmed that adsorption occurred through physical adsorption, electrostatic attraction, and struvite crystallization. The higher adsorption capacity of the Mg/Al-modified SBC was attributed to the transformation of the main adsorption form from physical adsorption to chemical adsorption. Furthermore, saturated MgSBC-0.5(@Al) was found to be a promising soil amendment. Therefore, electro-assisted modification showed substantial potential in the synthesis of Mg/Al bimetallic-modified BC and effectively increased the adsorption capabilities of SBC toward $NH_4^+$–N and $PO_4^{3-}$–P.

## Supporting information

**S1 Fig.** (a) The effect of Mg-loading for BC on the adsorption of $NH_4^+$-N and $PO_4^3$-P and (b) the XRD patterns of MgSBC-0.1(@Al), MgSBC-0.25 (@Al), MgSBC-0.5(@Al), and MgSBC-1 (@Al).
(TIF)

**S2 Fig.** (a) The fitting of adsorption isotherms of $NH_4^+$-N and $PO_4^3$-P by SBC at 298K and (b) adsorption kinetics of $NH_4^+$-N and $PO_4^3$-P by SBC: pseudo first-order and pseudo second-order model fitting, and (c-d) The effect of temperature on the adsorption capacity of $NH_4^+$-N and $PO_4^3$-P.
(TIF)

**S3 Fig. The removal efficiency of MgSBC-0.5(@Al) on total nitrogen and total phosphorus in actual biogas slurry.**
(TIF)

**S4 Fig. XRF spectrum of MgSBC-0.5(@Al)-NP.**
(TIF)

**S1 Graphical abstract.**
(TIF)

## Author Contributions

**Conceptualization:** Chu-Ya Wang.

**Data curation:** Qi Wang, Heng-Deng Zhou, Xiao-Lu Xiong.

**Formal analysis:** Qi Wang.

**Funding acquisition:** Chu-Ya Wang, Guangcan Zhu.

**Investigation:** Guangcan Zhu.

**Methodology:** Qi Wang.

**Project administration:** Qi Wang.

**Resources:** Chu-Ya Wang, Guangcan Zhu.

**Software:** Qi Wang, Dong-Xin Xue.

**Supervision:** Qi Wang, Chu-Ya Wang.

**Validation:** Qi Wang, Chu-Ya Wang.

**Visualization:** Qi Wang, Heng-Deng Zhou.

**Writing – original draft:** Qi Wang, Chu-Ya Wang.

**Writing – review & editing:** Qi Wang, Chu-Ya Wang.

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
