## [Decision Letter · Decision Letter 0]

12 Aug 2024

PONE-D-24-25807electro-assisted loading of magnesium/aluminium onto sludge-based biochar for the simultaneous adsorption of ammonia nitrogen and phosphatePLOS ONE

Dear Dr. Wang,

Thank you for submitting your manuscript to PLOS ONE. After careful consideration, we feel that it has merit but does not fully meet PLOS ONE’s publication criteria as it currently stands. Therefore, we invite you to submit a revised version of the manuscript that addresses the points raised during the review process.

We look forward to receiving your revised manuscript.

Kind regards,

G Yang

Academic Editor

PLOS ONE

Journal Requirements:

Chu-Ya Wang was supported by the natural science foundation of Jiangsu Province (BK20211047); Guangcan Zhu was supported by the natural science foundation of Jiangsu Province (BK20220038).

This work was supported by the Natural Science Foundation of Jiangsu Province (BK20211047 and BK20220038).

Chu-Ya Wang was supported by the natural science foundation of Jiangsu Province (BK20211047); Guangcan Zhu was supported by the natural science foundation of Jiangsu Province (BK20220038).

**Additional Editor Comments:**

Please 1. replace the keywords by biochar; ammonia adsorption; phosphate adsorption; adsorption mechanism; electro modification or others that are common are can be readily searched by others; 2. provide complete captions for Figure and Tables that can be understood with no need to read the text.

Reviewers' comments:

Reviewer's Responses to Questions

**Comments to the Author**

1. Is the manuscript technically sound, and do the data support the conclusions?

Reviewer #1: Partly

Reviewer #2: Yes

Reviewer #3: No

2. Has the statistical analysis been performed appropriately and rigorously? 

Reviewer #1: Yes

Reviewer #2: I Don't Know

Reviewer #3: No

3. Have the authors made all data underlying the findings in their manuscript fully available?

Reviewer #1: Yes

Reviewer #2: Yes

Reviewer #3: No

4. Is the manuscript presented in an intelligible fashion and written in standard English?

Reviewer #1: Yes

Reviewer #2: Yes

Reviewer #3: No

5. Review Comments to the Author

**Reviewer #1:** This study describes the production of a Mg/Al modified sludge-based biochar tested in the removal of phosphate and ammonia nitrogen from water. Despite some interesting results, several aspects should be improved and cleaned before a final decision. Specific comments are given below:

1. On page 13, line 3 of the ‘Batch adsorption experiments’ section, author mention that 'Mix a solution containing 50 mg·L-1 of PO4 3- -P and 100 mg·L-1 of NH4 + -N.' How did the authors determine the concentrations of the two pollutants and is there a correlation between the ratios of nitrogen and phosphorus? Please give the reasoning.

2. Why did author use H2SO4 for acidity adjustment in the 'Preparation of materials' section, but why use HCl for acidity adjustment in the 'Batch adsorption experiments' section?

3. In the section 'Batch adsorption experiments', 'Kinetic, isothermal and thermodynamic experiments', author do not give the experimental conditions such as pH value.

4. In the section 'Effect of different factors on NH4+-N and PO43--P capture', page 30, line 16, there is an error in 'Fe4(PO4)4'.

5. Page 36, 'Nitrogen and phosphorus recovery' section, lines 11-12, author suggest that 'Group A might promote the growth and development of mung bean sprouts because it absorbed rich N and P.' However, the ability of MgSBC-0.5(@Al)-NP to release N and P has not been studied in the manuscript, there is a lack of evidence to support this. It is suggested to supplement the related research on the performance of MgSBC-0.5(@Al)-NP in releasing N and P for a combined supplementary explanation.

6. Do the Mg/Al ions in MgSBC-0.5(@Al) leach? How about the release? This can be discussed in conjunction with the corresponding emission standards.

**Reviewer #2:** This study prepared biochar for simultaneous removal of ammonium nitrogen and phosphate. To help the authors improve this manuscript, specific suggestions and comments are put forward as follows:

1. Please re arrange the introduction. For example; you should mentioned kind of materials used to make biochar, then explain what is biochar.

2. Is that enough to remove moisture content at 105 C for 2 hour?

3. Do you measure soil characteristics? Please add in the manuscript

4. Germination rate calculation? please add in the manuscript

5. Soil experiments under room temperature or constant temperature? Under greenhouse or field experiment? Please make a detail, because next researcher need to understand the methodology and developed it.

6. I think the title should be change by adding its application on plant

**Reviewer #3:** The manuscript entitled “Electro-assisted loading of magnesium/aluminium onto sludge-based biochar for the simultaneous adsorption of ammonia nitrogen and phosphate” by Qi Wang et al. presents the modification of sewage sludge with Mg and Al through electrochemical methods. The Mg/Al modified sewage sludge is then pyrolyzed and the modified biochar is used for simultaneous adsorption of ammonium and phosphate. The premise of the study is quite unique because electrochemical modification is not commonly used for biochar modification. The manuscript has several major drawbacks:

(i) The mechanism of the electrically assisted impregnation is not clearly explained. To what extent does the impregnation method affect the biochar properties, and what is the mechanism?

(ii) The authors have prepared only two biochars (one with Mg/Al and one without). There is no variation of the impregnation conditions (e.g., impregnation time, current density) and pyrolysis conditions (e.g., pyrolysis temperature). The authors did vary the Mg acetate concentration, but those biochars were only tested in the adsorption experiment and not characterized. Thus, the material preparation conditions and the adsorption performance cannot be clearly related.

(iii) The role of aluminum is not clear. What was the role of Al in the adsorption of ammonium and phosphate? Or did the Al have a different role?

(iv) The main adsorption mechanism of ammonium and phosphate is explained to be due to struvite crystallization and formation of MgHPO4 and CaHPO4, but insufficient evidence is given. Such claims should be supported by a more in-depth characterization of the post-adsorption material (e.g., XRD measurements) supported by solubility calculations.

(v) A plant study was conducted to test the performance of the post-adsorption biochar as a soil amendment. This experiment has several problems namely (1) the number of replicates is not stated, (2) no positive control group was included, (3) no statistical analysis was carried out, and (4) the N and P contents in the sprouts were not measured.

(vi) Presentation and discussion of the results is confusing, both due to the language used and conclusions drawn based on the presented (insufficient) data.

More specific comments to improve the manuscript are given below:

1. Introduction: “Magnesium (Mg)/aluminum (@Al) bilayer metal … adsorption capability of BC”. Please provide a reference to this study.

2. Introduction: “When PO43--P, Mg2+, and NH4+-N are combined in solution, NH4+-N and PO43--P can be removed.”. Why is this? Please clarify.

3. Introduction: “Therefore, sludge has unique advantages over other BC materials in practical application [11].”. It is not clear how the authors come to this conclusion based on the information given in the discussed paragraph. Please clarify.

4. Materials and methods, preparation of materials: please explain where and how the sludge was collected.

5. Materials and methods, preparation of materials: “Subsequently, the initial pH was modified to 3.0”. Why was this done? Please clarify.

6. Materials and methods, preparation of materials: the experimental setup of the electrochemical modification is not clear. What was the anode material, and what was the cathode? Was a reference electrode used? Please clarify.

7. Materials and methods, preparation of materials: how were the NH4+-N and PO43—P solutions prepared?

8. Materials and methods, preparation of materials: please provide the purity and supplier of all reagents used.

9. Materials and methods, batch adsorption experiments: the authors have carried out the adsorption experiments using a solution containing 50 mg·L-1 of PO43--P and 100 mg·L-1 of NH4+-N. How/why were these concentrations selected? Please clarify.

10. Please refer to all equations in the text.

11. Please include the units of all symbols used in the equations.

12. Materials and methods, batch adsorption experiments: “At the temperature of 35 °C, for the pH experiment”. Why was a temperature of 35 °C used for the pH experiment, while 25 °C was used for the previous experimentss?

13. Equation 11: what is ‘D’? Please clarify.

14. Materials and methods, pot experiments: here the authors mention both MgSBC-0.5(@Al) and MgSBC-0.5(@Al)-NP. What is the difference between these materials?

15. Materials and methods, pot experiments: what is the “leaching process” that the authors are referring to here?

16. Materials and methods, pot experiments: the number of replicates in this experiment was not clear. Was only one pot used for each condition? Please clarify.

17. Materials and methods, pot experiments: in each pot experiment both a negative control and positive control should be used. How did the performance of the MgSBC-0.5(@Al)-NP compare to a commercial fertilizer? And what was the performance of the raw sludge (SP) in the mung bean growth experiment?

18. Materials and methods, characterization and analytical method: how were the samples prepared for ICP-MS measurements? Please clarify.

19. Results and Discussion, characterization and morphology: “Fig 1d shows the energy spectrum of MgSBC-0.5(@Al),demonstrating that the nano flake mainly consisted of Mg, Al, Si, chlorine (Cl), and oxygen (O) [12]”.

- What “nano flake” are the authors referring to here?

- Which sample does Fig. 1d refer to?

- What was the carbon content in the EDX analysis? Given that the authors have prepared biochars, the material is expected to contain a substantial amount of carbon.

20. Results and Discussion, characterization and morphology: “indicating that the electrical assisted modification could increase the SBET of BC, thereby increasing the adsorption capacities of pollutants.” A higher SBET may not necessarily lead to a higher adsorption capacity, because the SBET is not the only factor that affects the adsorption process. In various situations, materials with a low SBET have reached outstanding adsorption capacities because of the unique functional groups. The second part of the statement (“thereby…”) should be removed.

21. The number of significant figures should be checked and reduced in all tables. For example in Table 1: the measured pore volumes are not known with such precision.

22. Results and Discussion, characterization and morphology: “The peak observed at a wavelength of 465 cm-1 in SBC attributed to the Si-O-Si vibration. The peak at 465cm-1 corresponded to the SiO-Si or Al-O-Al vibration.” These two sentences can be combined into a single sentence.

23. Results and Discussion, characterization and morphology: “The results of FTIR further confirmed the existence of Mg and Al in MgSBC-0.5(@Al)”. This is not correct. The FTIR analysis cannot provide any information about the elemental composition. Please remove this statement.

24. Results and Discussion, adsorption kinetics and isotherm: at the beginning of this paragraph, the authors explain that they have prepared sludge-based biochars with varying Mg acetate solution concentrations. However, no explanation of these materials is given in the Materials and methods, and these materials have not been characterized. Please explain their preparation in the Materials and methods section, and provide a characterization of these biochars with the same techniques as MgSBC-0.5(@Al). This may provide insight into how the Mg acetate concentration affects the NH4+-N and PO43—P adsorption capacity.

25. Results and Discussion, adsorption kinetics and isotherm: “However, a high concentration of Mg2+ could cause the transformation of struvite crystals into Mg3(PO4)2.” Please provide experimental evidence for this statement or otherwise remove it.

26. Table 5: please mention the adsorbent in the caption.

27. Results and Discussion, Effect of different factors on NH4+-N and PO43--P capture: “The pHpzc values of MgO, AlOOH and Al2O3 were 12.0, 8.2 and 9.1, respectively [39, 40].” It is not clear why this is relevant here. Instead, what was the pHPZC of the adsorbents used in this study?

28. Results and Discussion, Effect of different factors on NH4+-N and PO43--P capture: “Simultaneously, the loss of NH4+ in the form of NH3 gas occurred at elevated pH levels. This led to a little decrease in the adsorption of NH4+-N”.

- What does “elevated pH levels” mean? Please specify the values.

- How does loss of N via NH3 gas decrease the adsorption capacity? If NH3 gas would escape, wouldn’t this lead to an overestimation of the adsorption capacity?

29. Results and Discussion, Effect of different factors on NH4+-N and PO43--P capture: “The existence of K+ and Na+ in the solution would compete with NH4+-N to form struvite potassium (KMgPO4·7H2O) and sodium magnesium PO43--P heptahydrate [NaMgPO4·7H2O].” In theory this is possible, but the solubility of K-struvite and Na-struvite may not be the same as that of regular struvite (depending on the pH). Please check the solubility of K-struvite, Na-struvite, and regular struvite at the pH of the experimental conditions used.

30. Fig. 5a: What was the pH of the solutions after the adsorption experiments?

31. Fig. 6: did the XPS signal of Al change after the adsorption experiment?

32. Results and Discussion, Nitrogen and phosphorus recovery:

- Were the treatments significantly different? Please include a statistical analysis.

- In Fig. 7 please include error bars for the wet weight

- Please include dry weight data.

33. Results and Discussion, Nitrogen and phosphorus recovery: “Therefore, MgSBC-0.5(@Al)-NP had the potential to replace N and P fertilizers.” This is not true, because no comparison is made with traditional N and P fertilizers. This statement should be removed.

34. Results and Discussion, Nitrogen and phosphorus recovery: what were the N and P contents in the seedlings?

35. The authors mention several times that adsorption of ammonium and phosphate was due to struvite formation. However, no evidence is given on the formation of struvite. X-ray diffraction analysis of the biochars after adsorption should be carried out to confirm whether any struvite was formed.

6. PLOS authors have the option to publish the peer review history of their article (what does this mean?). If published, this will include your full peer review and any attached files.

Reviewer #1: No

Reviewer #2: No

Reviewer #3: No

---

## [Author Response · Author response to Decision Letter 0]

12 Sep 2024

Response to Reviewer 1’s comments

This study describes the production of a Mg/Al modified sludge-based biochar tested in the removal of phosphate and ammonia nitrogen from water. Despite some interesting results, several aspects should be improved and cleaned before a final decision. Specific comments are given below:

1. On page 13, line 3 of the ‘Batch adsorption experiments’ section, author mention that 'Mix a solution containing 50 mg·L-1 of PO43--P and 100 mg·L-1 of NH4+-N.' How did the authors determine the concentrations of the two pollutants and is there a correlation between the ratios of nitrogen and phosphorus? Please give the reasoning.

Thank you for your question. The reason why we mixed 100 mg·L-1 NH4+-N solution and 50 mg·L-1 PO43--P solution in the experiment is: At present, the research group undertook the project of “transformation and promotion of low-carbon sub-quality treatment technology of Rural domestic sewage in Water network/Hilly Area” in Jiangsu Province. In some rural areas of Jiangsu Province, the concentrations of NH4+-N and total phosphorus in the effluent of septic tanks were generally around 100 mg·L-1 and 40-60 mg·L-1. In order to better simulated the removal of NH4+-N and phosphorus in actual black water and promoted the sewage treatment of the project to achieve better results, the initial concentration of this experiment was selected in this way.

2.Why did author use H2SO4 for acidity adjustment in the 'Preparation of materials' section, but why use HCl for acidity adjustment in the 'Batch adsorption experiments' section?

Thank you for your suggestion. We checked this area and found that it was a writing error, all of which were pH adjusted using HCl and NaOH.

3.

3.In the section 'Batch adsorption experiments', 'Kinetic, isothermal and thermodynamic experiments', author do not give the experimental conditions such as pH value.

Thank you for your suggestion. We added relevant conditions and revised this paragraph to:

“Modified BC (0.02 g) was added to 30 mL of the adsorption solution, the pH was adjusted to 7, and the mixture was shaken at a speed of 150 rpm.”

“Adsorption experiments were conducted at 25°C, 35°C, and 45°C, and the pH of the solution was adjusted to 7 using HCl and NaOH.”

“Adsorption kinetics were also investigated. Specifically, 50 mg·L−1 PO43−–P and 100 mg·L−1 NH4+–N solutions were mixed and stirred at 35°C and 150 rpm. The pH of the solution was adjusted to 7.”

4.In the section 'Effect of different factors on NH4+-N and PO43--P capture', page 30, line 16, there is an error in 'Fe4(PO4)4'.

Thank you for your suggestion. We changed this sentence to:

“Ca2+ and Fe3+ in the solution would immediately react with PO43−–P to form amorphous CaCO3 and FePO4, respectively, producing irregular crystals and seriously hindering the crystallization of struvite.”

5.Page 36, 'Nitrogen and phosphorus recovery' section, lines 11-12, author suggest that 'Group A might promote the growth and development of mung bean sprouts because it absorbed rich N and P.' However, the ability of MgSBC-0.5(@Al)-NP to release N and P has not been studied in the manuscript, there is a lack of evidence to support this. It is suggested to supplement the related research on the performance of MgSBC-0.5(@Al)-NP in releasing N and P for a combined supplementary explanation.

Thank you for your suggestion. We added the leading experiment to study the NH4+-N and PO43--P release performance of MgSBC-0.5(@Al)-NP. The relevant contents are as follows:

Fig 8. (d) NH4+-N and PO43--P leaching rates of MgSBC-0.5(@Al)-NP and commercial fertilizers.

The leaching characteristics of MgSBC-0.5(@Al)–NP were different from those of commercial fertilizers. The release of NH4+–N and PO43−–P in MgSBC-0.5(@Al)-NP showed a gradually increasing trend through the leaching process. When the L/S ratio reached 2.0 L·kg-1, the cumulative leaching amounts of NH4+–N and PO43−–P were only 2.53 mg·g-1(6.98%) and 6.05 mg/g (8.21%). At the end of the experiment, the cumulative leaching amounts of NH4+–N and PO43−–P were 11.54 mg·g-1(32.12%)and 26.36 mg·g-1 (35.78%), which were 2.72 times and 2.54 times lower than those of commercial fertilizers, respectively.

6. Do the Mg/Al ions in MgSBC-0.5(@Al) leach? How about the release? This can be discussed in conjunction with the corresponding emission standards.

Thank you for your advice. pH is the main factor affecting ion leaching, so we analyzed the leaching behavior of Mg/Al ions of MgSBC-0.5(@Al) elements in the pH range of 3~11. The intermittent leaching technique was used for experimental analysis according to the methods of the United States Environmental Protection Agency (USEPA 2014). The leaching concentrations of Mg2+ and Al3+ in eluent were analyzed by ICP-MS. When pH=3, the leaching amounts of Mg2+ and Al2+ were the highest, at 1.21 mg/L and 0.02 mg/L respectively. The state had no clear requirements for Mg2+ in groundwater, the evaluation was made according to the general content of tap water (magnesium content <300 ppm), which met the requirements of tap water. The amount of Al3+ leaching met the Chinese groundwater discharge Class II standard (< 0.05 mg/L).

Response to Reviewer 2’s comments

This study prepared biochar for simultaneous removal of ammonium nitrogen and phosphate. To help the authors improve this manuscript, specific suggestions and comments are put forward as follows:

1. Please re arrange the introduction. For example; you should mentioned kind of materials used to make biochar, then explain what is biochar.

Thank you for your suggestion. We re arranged the content of the introduction section. The overall writing idea is that: the current water pollution caused by PO43--P and NH4+-N. Definition and raw materials of biochar; Common modification methods and their drawbacks; The advantages and problems of electro-assisted modification method; The direction and content of our research. The second paragraph of the introduction section reads as follows:

“Biochar (BC) is a porous, loose, aromatic solid product formed through the pyrolysis of biomass under limited oxygen conditions. BC has a certain amount of surface functional groups, large specific surface area, wide availability of required raw materials, and low cost. Additionally, BC can adsorb pollutants from water [4]. Agricultural waste (such as straw, sawdust, sugarcane bagasse, and rice bran) and animal manure can be used as raw materials for the production of BC [5]. Some studies have shown that BC prepared from sludge exhibits strong adsorption capacity for organic and metal pollutants. Pyrolysis substantially reduces the ecological toxicity of heavy metals in sludge BC, reducing the environmental risks of its application [6]. Compared to BCs based on other types of biomass, sludge BC can be used to cost-effectively adsorb N and P. Therefore, sludge BC has unique advantages over other BC materials in practical applications [7]. However, the functionality and adsorption capacity of BC prepared via conventional pyrolysis are limited, making it unsuitable for pollutant adsorption. Consequently, BC is chemically modified, including impregnation modification with single metals (Fe, Mg, etc.) [8-10] and pairs of metals (Mg/Al, Ca/Mg, etc.) [11, 12]. Among the modified BCs, magnesium salt–modified biochar has high surface activity, anion fixation ability, and ion exchange ability. Magnesium-modified BC (Mg–BC) has been previously studied as a potential adsorbent for N and P. Mg–BC is prepared mainly through impregnation carbonization and carbonization impregnation. The modification of BC with Mg has been performed by soaking it in MgCl2 solution. However, the immersion time is long, usually more than 2 h [13]. Therefore, some researchers have proposed electro-assisted modification. Mg/Al bilayer metal–modified BC was prepared via an electro-assisted method using MgCl2 solution electrolyte and Al electrode, with an impregnation time of only 5 min. The formed compounds (MgO, spinel MgAl2O4, AlOOH, and Al2O3) evenly covered the surface of BC, exhibiting a highly organized and well-defined structure, resulting in improved PO43−–P adsorption capability of BC.”

2. Is that enough to remove moisture content at 105 ℃ for 2 hour?

Thank you for your question. After verification, due to the low water content of the sludge we selected, it could be dried for about 2-3 hours at 105 ℃. In order to be accurate and convenient for subsequent researchers, we changed the original text to: “After anaerobic fermentation, the sludge was dried in an oven at 105°C to remove moisture content, ground to a fine powder (smaller than 200 mesh), and stored in a dry place.”

3. Do you measure soil characteristics? Please add in the manuscript

Thank you for your suggestion. We tested the basic properties of the original soil and supplemented it in the text

Table 1. Basic properties of soil.

 Moisture content pH CEC organic matter Total phosphorus Total nitrogen Available Phosphorus Hydrolyzable nitrogen

Primitive soil % (cmol·kg-1) (g·kg-1) (g·kg-1) (g·kg-1) (mg·kg-1) (mg·kg-1)

 11.7 7.72 18.86 18.9 0.559 0.882 24.20 49

4. Germination rate calculation? please add in the manuscript

The germination rate (GP; %) of mung bean sprouts was calculated using Eq. (12).

 （12）

where n is the number of seeds germinated on the fourteenth day, and N is the total number of seeds.

5. Soil experiments under room temperature or constant temperature? Under greenhouse or field experiment? Please make a detail, because next researcher need to understand the methodology and developed it.

Thank you for your question. Our soil experiment was conducted under constant temperature in a greenhouse (with a temperature maintained at around 25 °C). The corresponding section in the article has been modified as follows：

“Five mung beans were planted in each pot and allowed to grow in a greenhouse at a temperature of around 25°C. Each grid was watered with 2.5 mL of tap water every day.”

6. I think the title should be change by adding its application on plant

Thank you for your suggestion. We revised the title as follows:

“Simultaneous adsorption of ammonia nitrogen and phosphate phosphorus on electro-assisted magnesium/aluminum-loaded sludge-based biochar and its utilization as a plant fertilizer”

Response to Reviewer 3’s comments

The manuscript entitled “Electro-assisted loading of magnesium/aluminium onto sludge-based biochar for the simultaneous adsorption of ammonia nitrogen and phosphate” by Qi Wang et al. presents the modification of sewage sludge with Mg and Al through electrochemical methods. The Mg/Al modified sewage sludge is then pyrolyzed and the modified biochar is used for simultaneous adsorption of ammonium and phosphate. The premise of the study is quite unique because electrochemical modification is not commonly used for biochar modification. The manuscript has several major drawbacks:

(i) The mechanism of the electrically assisted impregnation is not clearly explained. To what extent does the impregnation method affect the biochar properties, and what is the mechanism?

Thank you for your question. We found relevant literature and provided explanations, as well as supplemented the mechanism explanation section.

Through electrically assisted impregnation, the porosity of biochar could be effectively improved by oxidizing strong oxidants (providing more chemical binding sites) in about 10 minutes. In this experiment, Mg2+ and Al3+ could be loaded onto biochar to enhance its physical and chemical adsorption capabilities.

The adsorption capacities of NH4+–N and PO43−–P on biochar (MgSBC-0.5(@Al) impregnated with electrically assistance at 298k were 65.19 mg·L-1 and 92.10 mg·L-1, respectively. The adsorption capacities for NH4+–N and PO43−–P were 5.92 and 6.28 times than those of biochar without impregnation modification (SBC).

(ii) The authors have prepared only two biochars (one with Mg/Al and one without). There is no variation of the impregnation conditions (e.g., impregnation time, current density) and pyrolysis conditions (e.g., pyrolysis temperature). The authors did vary the Mg acetate concentration, but those biochars were only tested in the adsorption experiment and not characterized. Thus, the material preparation conditions and the adsorption performance cannot be clearly related.

Thank you for your suggestion. We supplemented the XRD characterization of biochar with different magnesium acetate concentrations and conducted correlation analysis.

Fig. S1 The XRD patterns of MgSBC-0.1(@Al), MgSBC-0.25 (@Al), MgSBC-0.5(@Al), and MgSBC-1(@Al)

“The four materials were characterized by XRD. It can be seen from S1b Fig that the peak of MgO in MgSBC-0.5(@Al) is higher than that of the other three materials, indicating that MgSBC-0.5(@Al) is loaded with more MgO.”

(iii) The role of aluminum is not clear. What was the role of Al in the adsorption of ammonium and phosphate? Or did the Al have a different role?

Thank you for your question. We reviewed the literature and found that the loading of Al3+could promote the enrichment of porous structures in biochar, which was beneficial for its application as an adsorbent. And some scholars have proven that metal ions (i.e. Al3+, Mg2+ and Ca2+) in the solution could also precipitate with PO43--P through strengthened chemical bonds, effectively improving the adsorption capacity of PO43--P. Therefore, the content and morphological properties of metal oxides (such as Al2O3) had a substantial impact on the adsorption capacity of PO43--P. We added the following explanation for the effect of Al3+in the text:

“The loading of Al2O3 was beneficial for the formation of porous structures in biochar. The presence of Al3+ in solution could precipitate with PO43--P through strengthened chemical bonds. This might improve the adsorption capacity of MgSBC-0.5(@Al) for PO43--P [20]”

(iv) The main adsorption mechanism of ammonium and phosphate is explained to be due to struvite crystallization and formation of MgHPO4 and CaHPO4, but insufficient evidence is given. Such claims should be supported by a more in-depth characterization of the post-adsorption material (e.g., XRD measurements) supported by solubility calculations.

We supplemented XRD characterization and found diffraction peaks of MgHPO4 at 2θ=27.81 ° and 18.51 °. However, due to the low content of CaHPO4, it could not be accurately identified in XRD. Furthermore, diffraction peaks of MgNH4PO4·6H2O were detected at 2θ=0.85 ° (111), 21.45 ° (021), 25.72 ° (200), and 30.19 ° (012), indicating the presence of MgNH4PO4·6H2O and MgHPO4 in MgSBC-0.5(@Al)-NP

Fig 6. (b) XRD spectra of MgSBC-0.5(@Al) adsorption of NH4+--N and PO43--P.

(v) A plant study was conducted to test the performance of the post-adsorption biochar as a soil amendment. This experiment has several problems namely (1) the number of replicates is not stated, (2) no positive control group was included, (3) no statistical analysis was carried out, and (4) the N and P contents in the sprouts were not measured.

(1)Each group of experiments was carried out four times, which had been added in the paper.

(2)Positive control group D (commercial fertilizer) was added.Fig 8. (a-b) Growth of mung beans on days 1 and 14, (c) Root length, stem length, average wet weight and average dry weight of five groups of mung beans on day 14, and (d) NH4+-N and PO43--P leaching rates of MgSBC-0.5(@Al)-NP and commercial fertilizers.

(3)We conducted a statistical analysis as follows.

-Thank you for your suggestions. We used SPSS statistics software to conduct substantially difference analysis on the four indicators, and the results are as follows,

After SPSS Statistics calculation, for stem length, average wet weight and average dry weight, there were substantially differences between groups A and E. And Group A was substantially different from the other three groups (Groups B, C, and D) (p<005). But for Root length, There was only a substantially difference between groups C and D. For mung bean sprouts, the application of commercial fertilizer (group D) had a great impact on the four indicators, and the effect was better than that of group A. But the application of group A (MgSBC-0.5(@Al)-NP) subs

---

## [Editor Report · Decision Letter 1]

18 Sep 2024

Simultaneous adsorption of ammonia nitrogen and phosphate phosphorus on electro-assisted magnesium/aluminum-loaded sludge-based biochar and its utilization as a plant fertilizer

PONE-D-24-25807R1

Dear Dr. Wang,

We’re pleased to inform you that your manuscript has been judged scientifically suitable for publication and will be formally accepted for publication once it meets all outstanding technical requirements.

Kind regards,

G Yang

Academic Editor

PLOS ONE
---

## [Editor Report · Acceptance letter]

16 Oct 2024

PONE-D-24-25807R1 

PLOS ONE

Dear Dr. Wang, 

I'm pleased to inform you that your manuscript has been deemed suitable for publication in PLOS ONE. Congratulations! Your manuscript is now being handed over to our production team.

Kind regards, 

on behalf of

Prof. G Yang 

Academic Editor

PLOS ONE